



# Inferring the seasonality of sea ice floes in the Weddell Sea using ICESat-2

Mukund Gupta[1,2], Heather Regan[3], YoungHyun Koo[4], Sean Minhui Tashi Chua[5,6], Xueke Li[7], and Petra Heil[5,6]

[1]Delft University of Technology, Delft, The Netherlands
[2]California Institute of Technology, Pasadena, USA
[3]Nansen Environmental and Remote Sensing Center, Bergen, Norway
[4]University of Texas at San Antonio, San Antonio, USA
[5]Australian Antarctic Division, Kingston, Australia
[6]Australian Antarctic Program Partnership, University of Tasmania, Hobart, Australia
[7]University of Pennsylvania, Philadelphia, USA

**Correspondence:** Mukund Gupta (Mukund.Gupta@tudelft.nl)

**Abstract.** Over the last decade, the Southern Ocean has experienced episodes of severe sea ice area decline. Abrupt events of sea ice loss are challenging to predict, in part due to incomplete understanding of processes occurring at the scale of individual ice floes. Here, we use high-resolution altimetry (ICESat-2) to quantify the seasonal life cycle of floes in the perennial sea ice pack of the Weddell Sea. The evolution of the floe chord distribution (FCD) shows an increase in the proportion of smaller

floes between November and February, which coincides with the asymmetric melt/freeze cycle of the pack. The freeboard ice thickness distribution (fITD) suggests mirrored seasonality between the western and southern sections of the Weddell Sea ice cover, with an increasing proportion of thicker floes between October and March in the south and the opposite in the west. There is a positive correlation between the mean chord length of floes and their average thickness, which persists throughout the year. Composited floe profiles reveal that smaller floes are more vertically round than larger floes, and that the mean roundness

of floes increases during the melt season. These results show that regional differences in ice concentration and type at larger scales occur in conjunction with different behaviors at the small scale. We therefore suggest that a comparison of floe-derived metrics obtained from altimetry could provide useful diagnostics for floe-resolving models and improve our understanding of sea ice processes across scales.

## 1 Introduction

Sea ice is a key resource of the climate system, providing shielding from solar radiation, regulating the ocean's overturning circulation (Gill, 1973), influencing lower latitude weather (England et al., 2020; Zhu et al., 2021), shaping interactions with ice shelves (Nicholls et al., 2009) and delivering critical support functions for polar ecology (Kohlbach et al., 2018; Vernet et al., 2019; Trathan et al., 2020; Fretwell et al., 2023). While Antarctic sea ice area has remained stable over the last few decades, the occurrence of several recent summer minima in the pack's extent (Parkinson, 2019; Turner et al., 2022; Purich and

Doddridge, 2023) may be signalling a longer term downward trend. The perennial extent of Antarctic sea ice is small compared



to the seasonal portion of the pack, but is particularly important for the polar ocean circulation and remains poorly understood, due to relatively sparse observations.

Over a seasonal cycle, the sea ice cover undergoes transformations that may provide insights into how the pack may respond to a changing environment. At the start of spring, solar heating initiates the melting of the pack, which facilitates the formation of cracks (or leads) due to stress induced by winds, ocean currents and waves (Rampal et al., 2009). This seasonal transition towards weaker ice does not always occur linearly with the drop in ice concentration, as storm events may break up the pack without a notable change in basal ice area (Hutchings et al., 2012). Further break up generates individual pieces of ice, or floes, which respond more readily to synoptic variability in the atmosphere and fine-scale turbulence in the ocean (Horvat et al., 2016; Brenner et al., 2023; Gupta and Thompson, 2022). The characteristics of sea ice floes also evolve during the winter months, due to dynamical processes including floe-floe collisions, ridging and rafting, which redistribute sea ice mass and area across scales (Hwang et al., 2017). Currently, due to the coarse resolution of climate models, the floe-scale connections between sea ice states across a seasonal cycle remain poorly understood and contribute to biases in sub-yearly predictions of sea ice extent (Bushuk et al., 2021).

The fine-scale properties of the sea ice pack have traditionally been characterized by the floe size distribution (FSD) inferred from imagery (Rothrock and Thorndike, 1984; Toyota et al., 2006; Denton and Timmermans, 2022). FSDs evolve seasonally, reflecting the proliferation of smaller floes in spring and summer and the consolidation of the pack during winter and fall (Steer et al., 2008; Geise et al., 2017; Stern et al., 2018). A cascade of processes governs this transition (Herman et al., 2021; Hwang and Wang, 2022), including fracturing of large floes ($> 500$ m) by ocean and atmosphere turbulence, grinding of medium-size floes (20 - 500 m) by waves and collisions, and lateral melt of small floes ($< 30$ m) due to oceanic heat (Steele, 1992). In winter, floes experience freezing, ridging and rafting (Timco and Burden, 1997; Vella and Wettlaufer, 2007; Herman, 2012), which congeal the ice back into large sheets. Theory-based inferences of sea ice fracture predict a variety of floe shapes and size distributions, but currently no single model can emulate the wide range statistics and phenomenological behaviors inferred from the observations (Herman et al., 2021; Montiel and Mokus, 2022), due to uncertainties regarding the modes of failure that sea ice experiences during break up and their applicability across scales. Due to the sparseness of polar imagery, inter-regional comparisons of floe-scale properties are also limited, and the relative importance of localized versus basin-wide forcings in governing sea ice dynamics remains uncertain.

Challenges in interpreting sea ice behavior also stem from the paucity of ice thickness data (Giles et al., 2008; Kurtz and Markus, 2012b), particularly in the Southern Hemisphere. Thicker ice is mechanically stronger (Hopkins, 1998) and can mitigate surface exchanges of heat, carbon, and light more efficiently than a thin cover (Stephens and Keeling, 2000; Gupta et al., 2020). Recent altimetric products, such as ICESat-2, provide high-resolution measurements of ice the freeboard ice thickness (Kwok et al., 2009; Laxon et al., 2013; Tilling et al., 2018; Kacimi and Kwok, 2020), which has been successfully leveraged to reduce sea ice volume biases in data-assimilated regional models (Fiedler et al., 2022; Williams et al., 2023). As with the FSD and FCD, the ice thickness distribution (ITD) inferred from altimetry follows predictable statistical laws, but its seasonality and regional variability remain understudied (Kwok et al., 2009). Early results from representing the joint floe size



and thickness distribution within climate models suggest that these fine-scale features can strongly influence the melt rate of the pack, but require tighter calibration to observational inferences to produce accurate predictions (Roach et al., 2018).

Our study focuses on the Weddell Sea, which hosts the most extensive perennial sea ice cover in the Southern Hemisphere (Eicken, 1992; Parkinson and Cavalieri, 2012). The physical conditions in the Weddell Sea are characterized by a large-scale gyral circulation, driven by synoptic winds, which advect sea ice in a clockwise motion across the basin to an outflow region

in the northwest. This circulation favors the accumulation of sea ice along the southern rim of the basin and the eastern edge of the Antarctic Peninsula (Hutchings et al., 2012), promoting substantial amounts of second-year sea ice in those regions. The mean thickness of this second-year sea ice typically exceeds 3 m (Haas et al., 2008), rendering it some of thickest sea ice in the Southern Ocean. The wind-driven export of sea ice away from the southern Weddell Sea also regularly exposes the sea surface to cold air, allowing the formation of new ice during a large fraction of the year. The seasonal melt and freeze cycle of sea ice,

combined with its gyre-driven advection, leads to complex multi-scale patterns in sea ice age, and thus properties, within the basin.

The Weddell Sea, typically known for its extensive winter sea ice, has become a focal point for the changing austral sea ice cover (Purich and Doddridge, 2023). While the pack is shielded by a clockwise gyre that traps sea ice, allowing it to persist and thicken, recent years have seen record lows in overall Antarctic sea ice extent. The freezing of sea ice, notably within

coastal polynyas, is responsible for the production of high-salinity shelf water, which will likely be sensitive to changes in the coverage of sea ice projected for the near future (Jeong et al., 2023). This suggests the Weddell Sea may not be immune to the effects of climate change, with potential ramifications for the entire Southern Ocean (Vernet et al., 2019).

This work uses the ICESat-2 altimeter product to examine the seasonality of the perennial sea ice zone in the Weddell Sea and explore the utility of fine-scale metrics in interpreting the basin-wide behavior of the pack. Section 2 details the datasets and

methodology used in the analysis, section 3 presents results regarding the seasonal evolution of sea ice types using microwave data and floe-scale properties using ICESat-2, section 4 provides a discussion of the results, and section 5 concludes.

## 2 Data products

### 2.1 Analysis regions

The analysis considers two distinct regions within the Weddell Sea, namely a western box (62 - 73 °S, 45 - 65 °W) and a

southern box (73 - 77.6 °S, 15 - 65 °W), as shown in Fig. 2 (a) and (b). These regions were selected because they contain a significant amount of sea ice throughout the year, which allows us to consider the full seasonal cycle of sea ice floes. The western region is in contact with the Antarctic Peninsula, though a portion of it extends beyond the tip of the peninsula, towards the Antarctic Circumpolar Current (ACC). The southern region is in contact with the Filchner–Ronne ice shelf in the south and the Brunt and Riiser-Larsen ice shelves in the east. The area of the western region is twice as large and has a stronger

north-south difference than the southern region, such that they capture different sea ice regimes.



## 2.2 Gridded sea ice products

We use the Multiyear Ice Concentration and Ice Type (MICIT) dataset (Shokr et al., 2008; Melsheimer et al., 2023), which uses passive brightness temperature and active microwave data to estimate daily concentrations of multiyear ice (MYI), first-year ice (FYI), young ice (YI), and open water at a nominal resolution of 12.5 km. We use the uncorrected version of the dataset, as the corrected version does not provide FYI and YI concentrations, though the timeseries of MYI area in the Weddell Sea compares well between the two products (not shown). Validation of the MICIT dataset with Sentinel-1 synthetic aperture radar (SAR) images, stage of development charts, and polynya data show acceptable accuracy and precision, especially for Antarctic MYI at the beginning of the freezing period (Melsheimer et al., 2023). We complement sea ice concentration data from MICIT with monthly gridded sea ice freeboard obtained from ICESat-2's ATL20 product at 25 km spatial resolution (Petty et al., 2020) and weekly gridded sea ice motion vectors at 25 km resolution from the Polar Pathfinder Version 4 dataset (Tschudi et al., 2019).

## 2.3 Along-track ICESat-2 altimetry

This study makes use of along-track data retrieved from the Ice, Cloud, and land Elevation Satellite (ICESat-2) laser altimeter (Markus et al., 2017; Neumann et al., 2019) between October 2018 and October 2022. We leverage sea ice freeboard measurements from the processed ATL10 product (version 6) (Kwok et al., 2019, 2021) in the western and southern regions, with a total of 3638 and 5996 tracks, respectively. We exclusively use the 3 sets of 'strong' beams data from each track, which are separated by 3 km in the across-track direction. The segment length is $\sim 15$ m and the individual laser footprint size is $\sim 11$ m, such that the effective along-track resolution is approximately $\sim 26$ m (Kwok et al., 2019; Petty et al., 2021). The freeboard measurements are provided relative to a local sea surface height obtained at sea ice leads within $\sim 10$ km of a given segment. These freeboard measurements are most reliable in areas of relatively high sea ice concentration ($> 50$ %, Petty et al. (2020)) , thus guiding our focus on the perennial ice cover.

We identify individual floes along a track according to their separation by 'specular' and 'dark' leads (specular leads: *height_segment_type* = 2–5; dark leads: *height_segment_type* = 6-9) (Kwok et al., 2022). Following past work (Horvat et al., 2019; Petty et al., 2021), we define the ice located between two consecutive leads along a track as a single floe, and the extent of that ice segment as the floe chord length (Figure 1). The mean freeboard of each ice floe is also calculated.

We note that several thick ice bodies, such as broken landfast ice or small icebergs, may remain along some ICESat-2 tracks, despite the iceberg filtering included in the ATL07/10 products (Kwok et al., 2022, 2023) (Figure 1c). These pieces of broken landfast ice and small icebergs typically have smoother and larger freeboard values, which can be visually distinguished from other sea ice floes. Here, we do not seek to precisely identify all these thick ice bodies in our tracks but rather estimate their approximate number to ascertain how they may affect our results. We first collect all the tracks comprising more than 5 floes whose mean freeboard height is greater than 1 m. We then manually count thick ice bodies within these tracks by visually identifying segments that have uncharacteristically smooth and high freeboard profiles (see Figure 1c). In the year 2019, we find that only 0.12 % of the detected sea ice floes (397 out of 314,582) are affected by the presence of thick bodies, such that these likely do not significantly impact the fITD results presented in section 3.





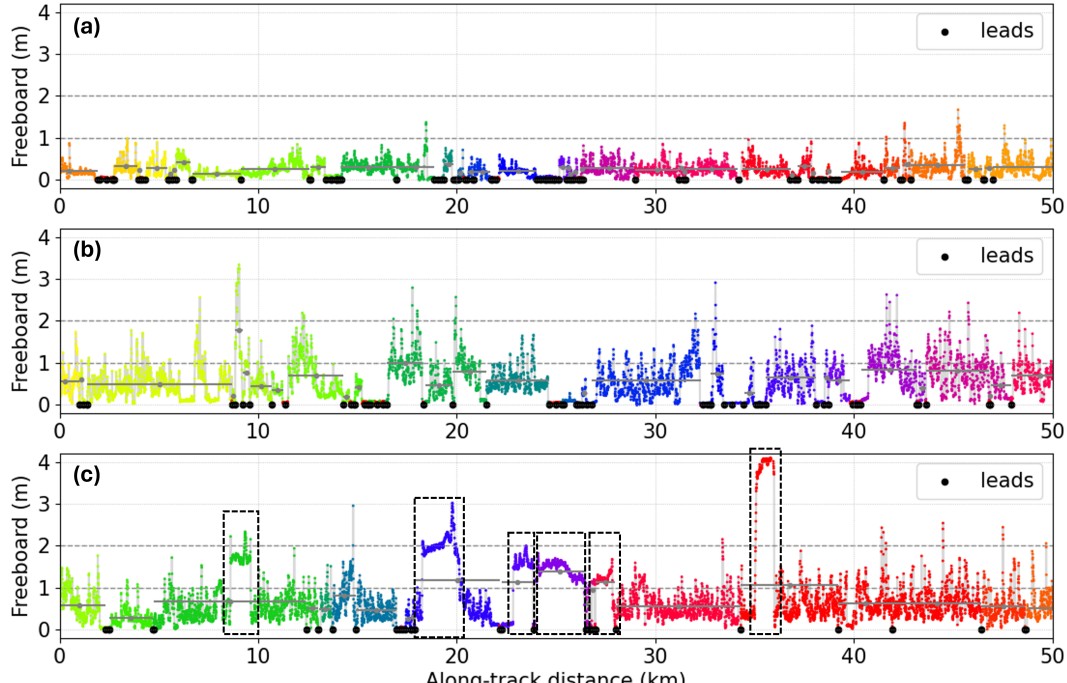

**Figure 1.** Identification of sea ice floes from along-track freeboard profiles of the ATL10 products taken on (a) March 9, 2019, (b) September 16, 2019, and (c) October 23, 2019. Black dots indicate the location of specular and dark leads detected by the ATL07/10 products. An ice segment between two consecutive leads is determined as a single floe, and different colors indicate individual floes identified based on the leads. Gray dots show the mean freeboard of individual floes, and gray vertical lines show the length of floes. In the track shown in panel (c), the segments displaying relatively smooth and high freeboard values likely indicate the presence of broken-up landfast ice (dashed boxes).

Given the lateral resolution of ICESat-2, the minimum floe size considered in this work is 25 m. We also note that the
ATL07/10 lead detection product does not always capture leads that are visible from concomitant Sentinel-2 imagery (Koo et al., 2023), which may cause an overestimate of floe sizes and an underestimate of the lead fraction. This relatively strict lead detection threshold can classify certain thin nilas as sea ice while others as open ocean. We test the sensitivity of our results to the following different lead definitions: (i) Specular + Dark (default), (ii) Specular, (iii) Freeboard threshold at 1 cm, and (iv) Freeboard threshold at 2 cm. Results pertaining to this sensitivity analysis are presented in the Appendix (Fig. A2 and A3).
While quantitative estimates of the floe chord distribution (FCD), lead width distribution (LWD), and vertical floe roundness (see section 3.4) can vary with these different lead detection algorithms, the broad conclusions of this work are not sensitive to these definitions.



## 3 Results

### 3.1 Sea ice types

The sea ice pack in the Weddell Sea is composed of various ice types, which may be broadly classified into young ice (YI), first-year ice (FYI) and multi-year ice (MYI) (Shokr et al., 2008; Ye et al., 2016b, a) (Fig. 2). The mean age of the ice tends to increase along the gyre's clockwise path (Lange and Eicken, 1991), with most of the YI on the eastern part of the basin, FYI in the central Weddell Sea, and MYI along the peninsula. The ocean circulation advects a fraction of YI toward the central gyre, where it interacts with FYI occupying most of the central and southern sections of the basin (Kacimi and Kwok, 2020).

We contrast the behavior of two separate regions in the Weddell Sea, the south and west (blue and yellow boxes in Figure 2), which together host the majority of the perennial sea ice pack, but exhibit different seasonality.

In the south, the total sea ice concentration remains relatively stable throughout the year, while the freeboard thickness displays significant seasonal variations (Fig. 2 (c)). The melt/freeze cycle is asymmetric, characterized by rapid melt between November and February, with approximately 15-30 % loss in freeboard thickness (except in 2020-2021), followed by steady

thickening during the rest of the year. The start of the freezing cycle in February is characterized by a peak in MYI, as most of the FYI remaining from the previous melt season converts to MYI. Between February and November, the freeboard thickness growth is accompanied by a drop in MYI, in favor of FYI, likely due to the clockwise advection of the gyre allowing for replenishment by local ice growth and advection of younger ice from the east.

In the west, the pack is mostly composed of MYI at the start of the freezing cycle (Fig. 2 (d)). Between February and

July, the MYI concentration remains stable and the total ice concentration increases due to growth and import of FYI from the East. Between June to August, a large fraction of MYI is advected out of the western box from its northern and eastern boundaries and replenished by a mixture of younger MYI and FYI from the southeast Melsheimer et al. (2023). Between July and December, the total sea ice concentration in the western region tends to drop, driven almost exclusively by a decline in the MYI concentration, as its supply from the south diminishes. Similarly to the southern box, the western region mostly comprises

thick FYI in November, but its mean freeboard thickness reduces by 40 - 65% during the melt season.

These results emphasize the importance of the basin-wide advection of sea ice in setting the spatial patterns of ice types within the Weddell Sea, particularly in compacting ice along the Peninsula and in driving MYI out of the perennial ice pack in favor of FYI. This circulation complicates correlations between sea ice age and thickness, notably as MYI exported to the north becomes more vulnerable to melt than younger ice in the south.



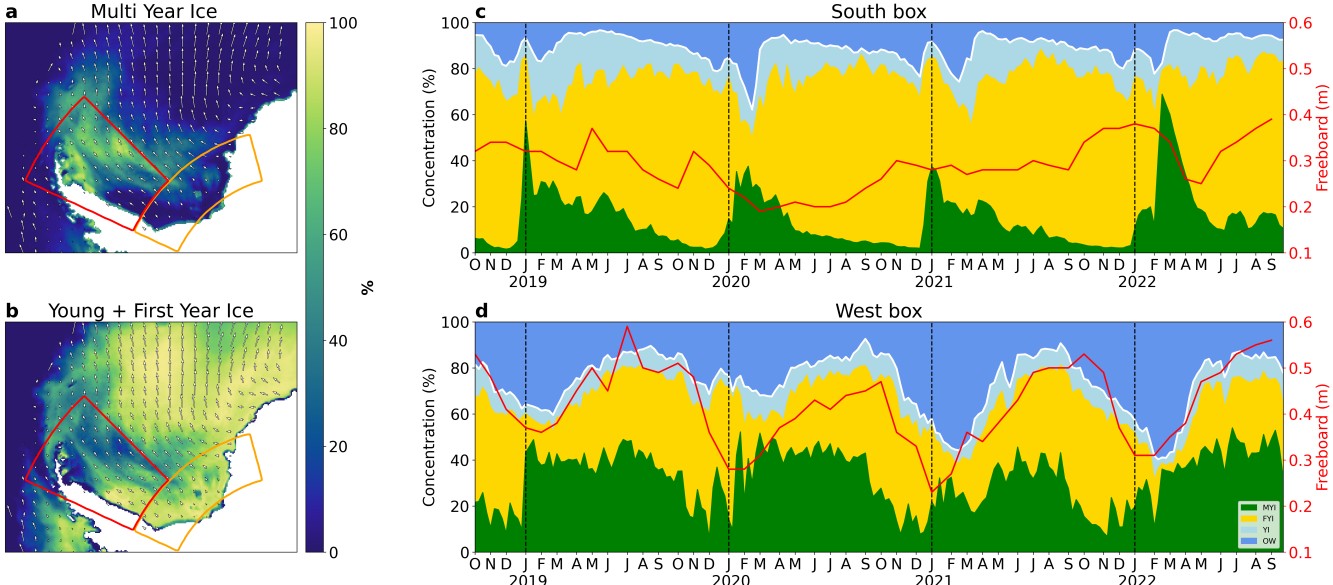

**Figure 2. Seasonality of sea ice types in the perennial ice pack.** (a-b) Sea ice concentration for (a) MYI and (b) YI+FYI in the Weddell Sea averaged between June-August 2019, with concurrent sea ice drift vectors (Melsheimer et al., 2023; Tschudi et al., 2019). The yellow and blue boxes in each panel delimit the southern and western regions, respectively. (c-d) Weekly-resolved time series of sea ice types averaged over the southern (c) and western (d) regions. The white line highlights the total sea ice concentration. The red line represents the area-weighted sea ice freeboard thickness at monthly resolution obtained from ICESat-2's gridded product (ATL20).

## 3.2 Floe chord length and freeboard thickness

The large-scale properties of the perennial ice pack differ between the southern and western parts of the Weddell Sea, and throughout the year (Fig. 2), motivating a more detailed investigation at finer scales. Here, we use ICESat-2 altimetry to examine the floe chord distribution (FCD), freeboard ice thickness distribution (fITD), lead width distribution (LWD), and the vertical profiles of floes across the seasonal cycles.

Averaged over the study period (October 2018 - October 2022), the FCD displays a monotonic decline from approximately 70 m to 10 km. This distribution can be interpreted as a higher proportion of small floes relative to larger ones, across the entire range of chord lengths considered (Fig. 3 (a)). We fit the FCD using a power law with exponent $\alpha_{FCD}$, between chord lengths spanning 100 m to 10 km and a uniform bin size of 50 m, which gives $\alpha_{FCD} = -1.2$ in both regions. Therefore, despite containing different sea ice types, both the southern and western regions have similar annual-mean FCD slopes.

The fITD is evaluated as the count of individual ICESat-2 segments of approximately 10 m length over thickness bins varying between 0.1 m to 1.5 m. The distribution declines mostly monotonically over these bin sizes, which signifies a larger proportion of thin segments relative to thicker ones, except for a small fITD peak at around 0.2 m in both regions (Fig. 3 (b)). We approximate the fITD as an exponentially decaying curve with a coefficient $\alpha_{fITD}$ between 0.2 m and 1.5 m and a bin size of 0.02 m. The fITD slope is relatively steep in the southern box ($\alpha_{fITD} = -1.7$) and shallower in the western box





($\alpha_{fITD} = -1.4$), reflecting the presence of thicker ice near the Antarctic Peninsula. In the west, the shallower fITD slope is consistent with the higher fraction of MYI throughout the seasonal cycle (Fig. 2).

We evaluate the joint chord length and freeboard thickness distribution using the same bin sizes as for the individual distributions (Fig. 3 (c)). The contours of the joint distribution have a parabolic shape, such that small and thin floes significantly outnumber thick and large ones (Fig. 3 (c)). Very high freeboard values ($> 1$ m) are also observed over small floe lengths,

which may be affected by the presence of icebergs, as discussed in Section 2.3.

The peak of each contour of the joint distribution represents the freeboard thickness of the majority of floes within a given floe chord length range. This peak also coincides with the mean freeboard thickness evaluated over that range, which is depicted by the dotted red line. The mean freeboard thickness increases with floe chord length, with a slope $\alpha_{CLF} = 0.04$, as evaluated from $h \propto \alpha_{CLF} log(x)$). Therefore, on average, larger floes have a greater freeboard thickness than smaller floes. This conclusion

also holds during individual months for both regions, as shown in Fig. A1. We note that while the mean floe thickness is positively correlated with the mean floe chord length, this correlation is negative for thicker floes, as shown by the downward-sloping contours of the joint distribution for large freeboard.

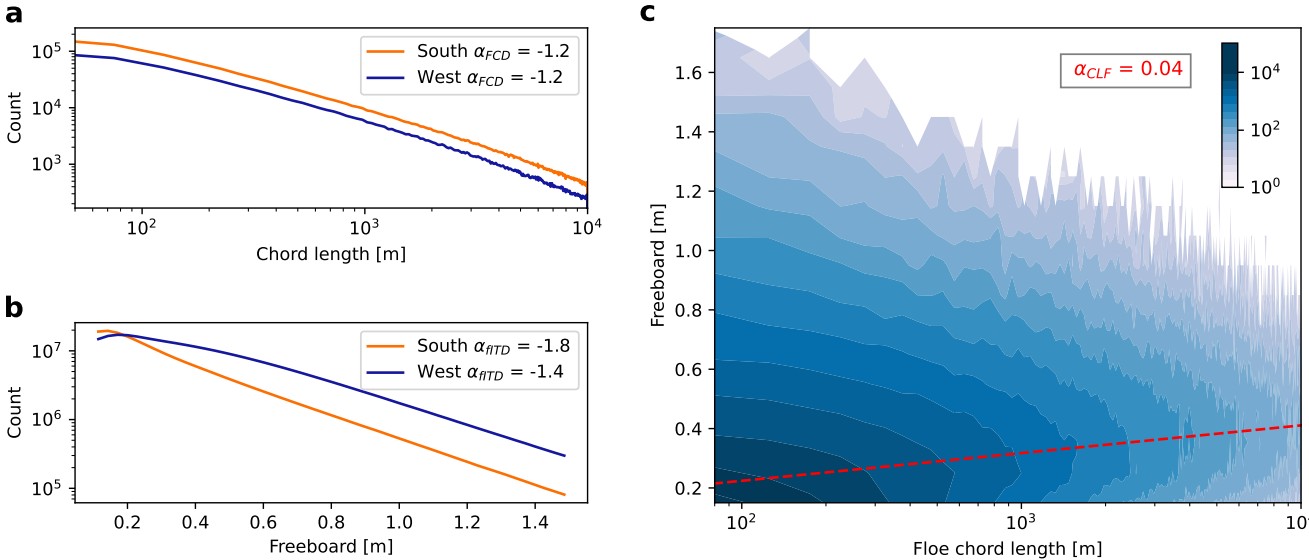

**Figure 3.** (a) FCD evaluated over 50 m bin size and (b) freeboard ice thickness distribution (fITD) evaluated over 0.02 m bin size for the south (orange) and west regions (blue), respectively. The best fit slope for each curve is indicated in the legend ($\alpha_{FCD}$ and $\alpha_{fITD}$, respectively). (c) Joint floe chord and freeboard thickness distribution showing the number of floes counted over the same bin ranges as in (a) and (b). The dotted red line marks the mean freeboard value over the chord length range, and has a slope of $\alpha_{CLF} = 0.04$ on this graph, with a Pearson correlation coefficient of $r^2 = 0.98$.

We investigate the variability of the FCD and fITD by evaluating these quantities over sub-samples of the total 4-year period. We choose to study a characteristic seasonal cycle by detrending and compositing the anomaly in the slope of these





distributions. We evaluate $\alpha_{FCD}$, $\alpha_{fITD}$ and $\alpha_{CLF}$ in chunks of 3 days, calculate the time series of the anomaly relative to the detrended annual mean, and composite the results over a seasonal cycle. The final time series is smoothed using a periodic Gaussian filter with a window of 6 days. We find that the seasonality of the slopes is not very sensitive to the number of chunks chosen to evaluate the distributions or the various lead definitions considered in this work (Fig. A2).

The southern and western regions of the pack display a consistent seasonal evolution in their FCD slope anomaly $\alpha'_{FCD}$ (Fig. 4 (a)). Around November, at the start of the melt season, $\alpha'_{FCD}$ is positive, which represents a higher proportion of larger floes relative to the annual mean. Throughout the melt season, between November and February, $\alpha'_{FCD}$ decreases and becomes negative around December, as a result of the pack fracturing into increasingly small floes. From February to October, $\alpha'_{FCD}$ increases and becomes positive again around April, as sea ice refreezes into a more consolidated pack composed of larger floes. Despite some inter-annual variability, $\alpha'_{FCD}$ is statistically different from zero for a sizeable portion of the year in both regions.

Unlike the FCD, the seasonality of the fITD slope suggests an anti-phase relationship between the southern and western regions of the sea ice cover (Fig. 4 (b)). In the west, $\alpha'_{fITD}$ is positive at the start of the melt season, which signifies a higher proportion of thick ice segments relative to the annual mean. During the melt season, between November and February, $\alpha'_{fITD}$ declines and becomes negative around December, such that the proportion of thicker ice becomes smaller than the annual mean. Between February and October, $\alpha'_{fITD}$ increases and becomes positive again around June. Despite some inter-annual variability, $\alpha'_{fITD}$ is statistically different from zero for a sizeable portion of the year in the west. In the south, around November, $\alpha'_{fITD}$ is slightly negative. During the melt season, between November and February, $\alpha'_{fITD}$ increases and becomes positive around January, such that the proportion of thicker floes becomes greater than in the annual mean. During the freezing season, between February and October, $\alpha'_{fITD}$ declines and reaches slightly negative values. However, inter-annual variability is large in the south, and the fITD signal is not statistically significant, except between mid-January to April.

The correlation slope between mean freeboard thickness and mean floe chord length ($\alpha_{CLF}$, red dotted line in Fig. 3 (c)) also varies seasonally (Fig. 4 (c)). The magnitude of these fluctuations is on the same order as the annual mean signal, such that the sign of the correlation between chord length and freeboard thickness remains positive throughout the year, as seen over individual seasons (Fig. A1). As with $\alpha'_{fITD}$, the composited-mean seasonal fluctuations in $\alpha'_{CLF}$ are generally mirrored between the south and west regions. However, inter-annual variability is large and the seasonal signal in $\alpha'_{CLF}$ is not statistically significant over any part of the year.




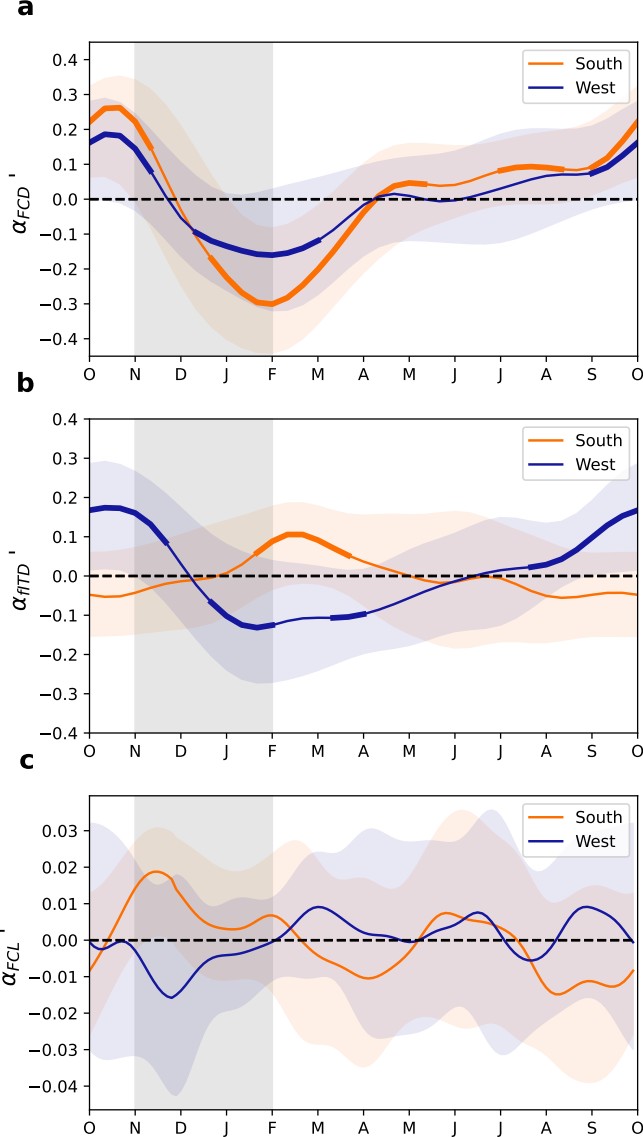

**Figure 4.** (a) FCD slope anomaly ($\alpha'_{FCD}$) composited over a seasonal cycle, and smoothed by a periodic Gaussian filter with a window of 6 days. Positive values of $\alpha'_{FCD}$ signifies a reduced proportion of small floes relative to the annual mean. The colored shadings represent two standard deviations inter-annual variability and the bold parts of the curve represent periods during which $\alpha'_{FCD}$ is significantly different from zero, based on a p-value of 0.05. The vertical grey shadings highlight the approximate duration of the melt period, as inferred from Fig 2. (b) As in (a) but for the fITD slope anomaly ($\alpha'_{fITD}$), where a positive anomaly signifies a reduced proportion of thin floes relative to the annual mean. (c) as in (a) but for the slope between mean floe chord length and freeboard thickness ($\alpha'_{CLF}$), where a positive anomaly signifies a increased positive correlation.



## 3.3 Lead width and spacing

The size and thickness of floes are influenced by the width and spacing between sea ice leads, which provide large heat exchanges with the ocean and atmosphere, as well as space for inter-floe collisions (Maykut and Perovich, 1987). We investigate the lead width distribution (LWD) by binning leads according to their size along individual tracks. As with the FCD, we focus on the dark and specular leads identified by the ATL07 algorithm, and report on the sensitivity of our results to the lead definition in Fig. A3.

We evaluate the LWD by aggregating the leads identified over the 4-year data collection period in bins with widths ranging from 20 m to 3 km. The LWD is almost identical for the southern and western regions of the pack, characterized by a mostly monotonic decrease with size, which represents a larger proportion of narrow leads compared to wide leads (Fig. 5 (a)). The variability in the LWD for lead widths greater than 700 m is caused by their relatively small count ($< 10$). For statistical significance, we do not take account of these wide leads in our subsequent analysis. The mean slope of the LWD evaluated between 20 - 700 m is $\alpha_{LWD} = -3.1$ for both regions, which is substantially steeper than $\alpha_{FCD}$ (Fig. 3).

Following the method used to produce the panels in Fig. 4, we evaluate the seasonal composite of the LWD slope anomaly $\alpha'_{LWD}$ in Fig. 5 (b). Over both regions, $\alpha'_{LWD}$ tends to be positive in October, which represents a larger fraction of wide leads relative to the annual mean. During the melt season, $\alpha'_{LWD}$ declines and becomes negative, representing a larger fraction of narrow leads relative to the annual mean. The LWD then progressively increases back to its October distribution over the course of the freezing period. The seasonal signal in $\alpha'_{LWD}$ is only statistically significant during parts of the melt season and towards the end of the freezing period. The magnitude of the anomalies tend to be larger in the south compared to the west. While there may be differences in the phasing of the signal between the two regions, those are difficult to evaluate, due to large inter-annual variability.

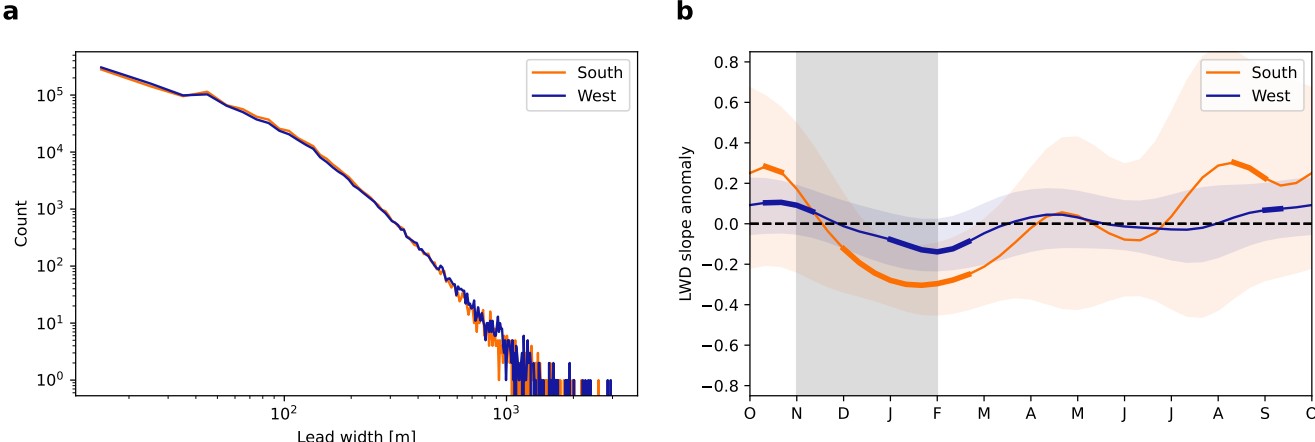

**Figure 5.** (a) Lead width distribution for the south (orange) and west (blue) regions evaluated over a bin size of 10 m. (b) As in Fig. 4 but for the LWD slope anomaly ($\alpha_{LWD}$). Positive values of $\alpha_{LWD}$ signify an increased proportion of wider leads relative to the annual mean.





The spacing between leads may also a useful metric for describing the compactness of the sea ice cover, and its potential response to atmospheric forcings and surface heat fluxes (Farrell et al., 2020). Here, we evaluate the mean spacing between leads, binned over lead widths ranging from 25 m to 300 m, with a constant bin spacing of 10 m (Fig. 6). We do not examine
the spacing between leads wider than 300 m, as they do not occur frequently enough to provide robust statistics (Fig. 5 (a)). In the annual mean, the average spacing between leads contained within the 50 m width bin is approximately 20 km for both regions. In the west, the mean lead spacing increases to a maximum value of 102 km for 180 m wide leads, while in the south the mean lead spacing increases to a maximum of only 41 km for 150 m wide leads. The lead spacing curves subsequently flatten out around their respective maxima in both regions. Consequently, the mean lead spacing is up to 2.5 times larger in the
west compared to the south, despite these two regions having the same $\alpha_{LWD}$ and $\alpha_{FCD}$.

The relationship between lead spacing and lead width becomes less clear with lead widths increasing from the maximum lead spacing. For example, beyond lead widths of 200 metres in the western region, the curve becomes increasingly interrupted. Though the maxima of each curve here occurs at a different lead width, there is a similar breakdown of the relationship as lead width increases beyond the maximum lead spacing. This is also the case with different lead definitions (Fig. A3). Additionally,
when using a centimeter threshold definition, the lead spacing curves decrease beyond the maxima rather than flattening. Thus, there appears to be a limit to how large a lead width can be while still having a discernible relationship with lead spacing.

We probe the seasonality of the lead spacing by aggregating data for individual months across the 4-year period (Fig. 6). In both regions, the broad shape of the lead spacing distribution is similar to the annual mean, but with varying magnitudes. In the west, from July to January, the mean spacing between leads decreases across all lead width categories, reflecting a
more fractured sea ice cover by the end of the melt season, before increasing again between April and July. In the south, the seasonality in mean lead spacing is less pronounced. Narrow leads ($< 120$ m) follow a similar phasing as in the west, while wider leads ($> 120$ m) have larger mean spacing in the summer compared to the winter. Characterizing leads using a freeboard threshold instead of the identification provided by ICESat-2 can reduce the seasonality in the lead width spacing signal (Fig. A3). This may be due to the misidentification of thin ice as leads, especially near areas of widely distributed thin ice (Koo
et al., 2023). Nevertheless, the seasonal trend in lead width spacing remains consistent across lead definitions.



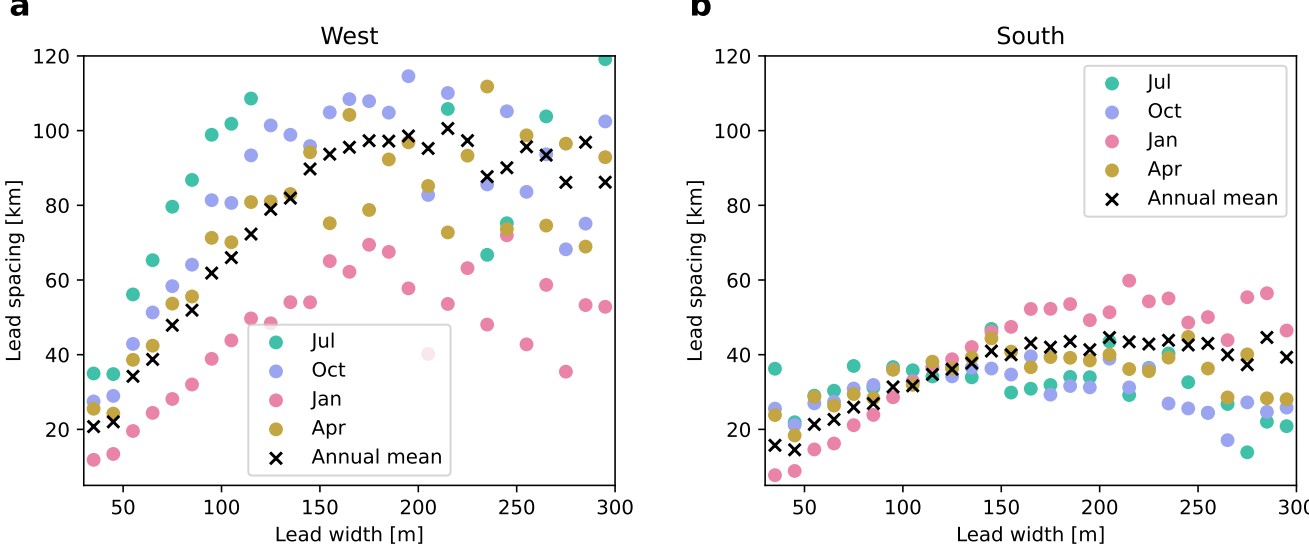

**Figure 6.** Mean lead spacing computed at each lead width bin category between 40 - 300 m for (a) the western and (b) the southern regions. The colored scatter points represent aggregates for selected months, while the black 'plus' and 'cross' symbols represent yearly-mean aggregates for the two regions, respectively.

### 3.4 Vertical floe roundness

The ICESat-2 along-track freeboard data can be used to investigate the shape of floes in the vertical direction and their associated changes over the seasonal cycle. We aim to quantify the characteristic vertical profile of floes by compositing their freeboard thickness along their respective chord lengths. The floes are grouped into the following three size categories for the
compositing process: small (100 - 500 m), medium (500 m - 5 km), and large (5 km - 50 km). Composites are produced by averaging the freeboard thickness over half the normalized floe chord distance, such that the resulting average represents the mean floe profile from the floe edge to the center of the chord length. Both halves of a given floe chord are weighed equally and considered independently in the compositing procedure.

The mean vertical profiles of freeboard thickness reveal a semi-dome shape for all three size categories (Fig. 7 (a)), such that
floes on average have smaller freeboard at their edges compared to their chord centers. While individual vertical profiles may show significant variability, the standard deviation associated with the mean profiles is negligibly small, due to the large number of floes considered in each size category ($O(10^5)$). Consistent with inferences obtained from the joint chord and freeboard thickness distribution, (Fig. 3 (c)), the mean composited profiles show that larger floes have a greater overall freeboard thickness than smaller floes.

The mean vertical shape of floes may be useful indicators of processes occurring at these scales. To characterize these floe shapes, we evaluate the distance $d$ over which the composited profiles become flat, based on a slope threshold of 0.05 in normalized units. The distance $d$ is 97 m for small floes, 456 m for medium and 1.3 km for large floes, or 0.78, 0.57, and 0.36



respectively in normalized units. The accuracy of these $d$ values may be affected by the altimeter footprint ($\sim$ 11 - 26 m), but this error is expected to diminish with the compositing procedure.

We further quantify the vertical roundness of floes $\alpha_{vr}$ by integrating under the curve of the mean floe profiles and normalizing by the area of a rectangle enclosing the profile, as follows:

$$\alpha_{vr} = 1 - \frac{\int_0^1 \hat{p}(\hat{x})\,d\hat{x} - \pi/4}{1 - \pi/4}, \tag{1}$$

such that a perfectly round profile exhibits $\alpha_{vr} = 1$ and a square profile tends to $\alpha_{vr} = 0$. In the annual mean, $\alpha_{vr}$ is 0.76 for small floes, 0.37 for medium floes and 0.21 for large floes. The smallest floes are thus 3.6 times rounder than the largest floes

on average.

In both regions, $\alpha_{vr}$ averaged across all floes varies over the seasonal cycle. In the south, $\alpha_{vr}$ tends to increase between August and February, before dropping sharply between March and April, at the start of the freezing cycle (Fig. 7 (b)). In the west, $\alpha_{vr}$ increases between August and September, but inter-annual variability is too large to conclusively report on the rest of the seasonal cycle. In both regions, $\alpha_{vr}$ averaged over individual size categories does not vary significantly throughout the year

(not shown), which means that the seasonal variability in the mean vertical roundness across all sizes are driven by changes in the size and freeboard thickness distribution of floes. The increase in $\alpha_{vr}$ during spring and summer is thus likely linked to the proliferation of smaller floes with rounder profiles. The timing of the seasonal cycle in $\alpha_{vr}$ is not highly sensitive to the different floe definitions considered in this work (not shown).

## 4   Discussion

The objective of this work was to develop and explore the utility of various fine-scale sea ice metrics in advancing our understanding of the seasonality of the pack within the Weddell Sea. Our investigation of coarse-resolution gridded datasets provides large-scale context for the evolution of sea ice concentration, types and freeboard thickness, while ICESat-2 altimetry provides inferences pertaining to individual floes and leads. Despite ongoing challenges in interpreting some of these observations, our results suggest that along-track and floe-resolving measurements can provide useful information on the physics of sea ice at

the basin scale.

The spatial patterns in sea ice types show that the western portion of the Weddell Sea is composed of thicker and older sea ice than in the south (Section 3.1), which is consistent with past inferences (Haas et al., 2008; Kacimi and Kwok, 2020; Melsheimer et al., 2023). The annual-mean sea ice concentration is also lower in the west due to summer melt and northward export towards the ACC. In spite of these regional differences, the seasonality of the FCD is consistent between the western and southern

portions of the pack and is in phase with the basin-wide asymmetric melt/freeze cycle (Section 3.2). The spatial homogeneity in FCD suggests that the size distribution of floes is not strongly sensitive to localized differences in environmental conditions, but may instead be controlled by processes occurring at the Weddell Sea basin scale. While the processes responsible for the



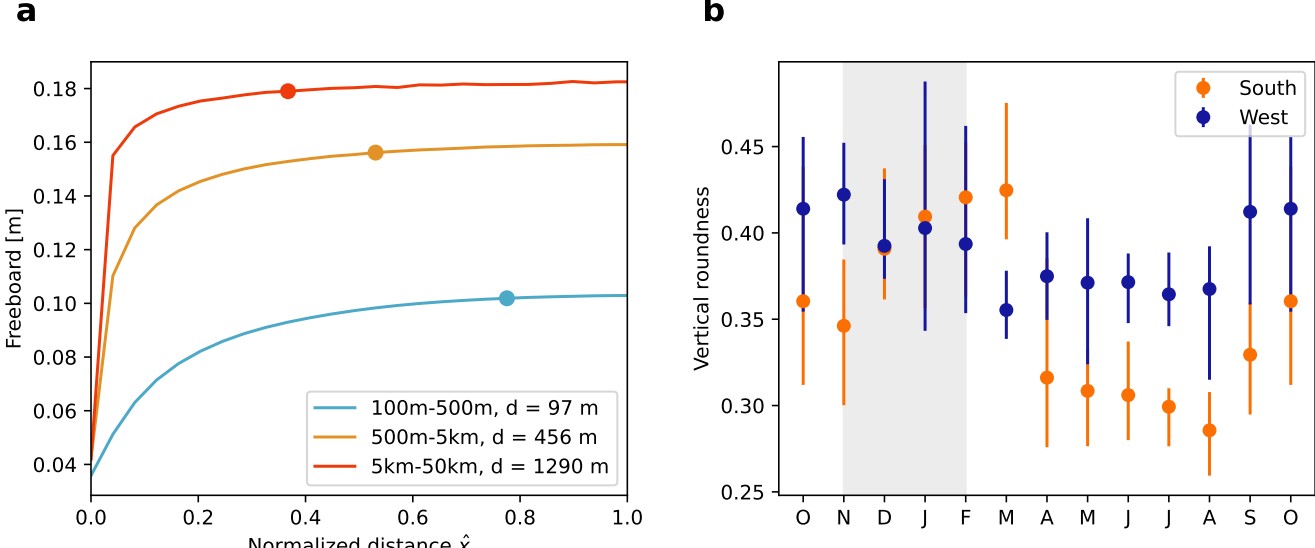

**Figure 7.** (a) Composited vertical floe profiles obtained by averaging the freeboard distribution along each floe and over a normalized horizontal distance $\hat{x}$. The compositing is done separately for small (100 - 500 m, blue), medium (500 m - 5 km, orange) and large (5 - 50 km, red) floes, with each category comprising $2.8 \cdot 10^5$, $3.6 \cdot 10^5$ and $1.5 \cdot 10^5$ floes, respectively. The colored dots represent the distance $d$ over which the profiles flatten, defined as the point along $\hat{x}$ where the profile slope drops below 0.05 in normalized units. (b) Monthly-mean vertical roundness of floes for the south (orange) and west (blue) regions calculated from floe profiles including all sizes (100 m - 50 km). The error bars represent the minimum to maximum ranges of vertical roundness reported for the 4-year ICESat-2 period. The vertical grey shading highlights the approximate duration of the melt period.

asymmetry in the melt/freeze cycle remain to be fully understood (Roach et al., 2022; Goosse et al., 2023), our results suggest that its phasing plays a first order role in controlling the FCD.

In contrast to the FCD, the freeboard fITD displays notable differences between the southern and western parts of the Weddell Sea (Fig. 4). In the annual mean, there is a greater fraction of thicker freeboard segments in the west, likely due to strong compaction against the Antarctic Peninsula (Yi et al., 2011). During the melt season, while the total freeboard thickness decreases in both regions (Fig. 2 (c-d)), the fraction of thicker ice segments increases in the south and decreases in the west (Fig. 4 (b)). The diminishing fraction of thick ice in the west is consistent with the observed advection of MYI away from the Antarctic Peninsula (Fig. 2 (a)), and potentially reduced compaction due to looser sea ice concentration. In the south, the increasing fraction of thick ice in summer may instead be caused by the preferential melt of thin ice in response to solar forcing or changes in the import of younger ice from the east. Seasonal changes in the freeboard thickness may also be modulated by snowfall, whose effect cannot be parsed from ICESat-2 data alone. At coarse scales, previous studies have employed linear relationships between snow depth and freeboard (Kacimi and Kwok, 2020; Kurtz and Markus, 2012a; Xie et al., 2011; Worby et al., 2008), but further observational is required to assess the relationship between the ITD and fITD at fine spatial scales



and over different parts of the Weddell Sea (Arndt, 2022). Furthermore, the variability in the fITD is large, such that a longer observation period will be required to better understand the role of the physical processes active over sub-seasonal to inter-annual time scales.

Metrics pertaining to sea ice leads provide additional information on the spatial structure of the pack. As with the FCD, the LWD follows a similar structure between the western and southern portions of the Weddell Sea (Fig. 5 (a)). On the other hand, the mean lead spacing, when calculated within individual lead width categories, is higher in the west by a factor of up to 2.5 relative to the south (Fig. 6). This mean lead spacing is not necessarily equivalent to the mean floe size, as it additionally depends on the sea ice concentration, which differs between the two regions, and the ordering of individual leads along a track. The larger lead spacing in the west is consistent with a more compacted pack due to sea ice impinging upon the Antarctic Peninsula (Lange and Eicken, 1991). The seasonal change in the lead spacing also coincides with increased compaction producing more sparsely distributed leads in the winter, and the opposite in the summer. Further statistical analysis of the spatial distribution of leads may therefore be an important metric in characterizing upcoming changes to the sea ice cover.

Throughout the seasonal cycle, the joint chord-freeboard distribution displays a positive correlation between the mean floe chord length and the mean floe freeboard inferred by ICESat-2 (Fig. 3 and 4 (c)). This finding is consistent with lateral melt preferentially affecting smaller floes in the summer (Steele, 1992; Horvat et al., 2016; Gupta and Thompson, 2022), but remains to be understood for the rest of the seasonal cycle. Processes such as snow export into leads (Leonard and Maksym, 2011; Moon et al., 2019), flooding (Ackley et al., 2020; Yi et al., 2011) and the formation of thin nilas at floe edges (Farrell et al., 2020) may contribute to this positive correlation, but further work would need to assess their relative importance. The composited vertical shapes of floes inferred from our analysis show that the characteristic vertical rounding distance $d$ increases with floe size (Fig. 7), which suggests that different erosive mechanisms may dominate at different spatial scales.

The parabolic shape of the joint chord-freeboard distribution shows that floes with the highest freeboard thickness values are found within the smallest floe chord length bins (Fig. 3). This is in contrast to the mean floe freeboard, which is positively correlated with chord length, such that smaller floes tend to be thinner on average. Simultaneously thick and small floes instead may be generated by a combination of ridging when two floes collide, and snow accumulation processes. Small icebergs may also be indistinguishable from sea ice floes in the ICESat-2 data, and could contribute to this category. Furthermore, the largest floes found by the altimeter have kilometer-scale chord lengths, but may in reality be composed of smaller floes that are frozen together by thinner ice. While more sensitive lead detection schemes can locally identify a greater number of smaller floes, we find that the seasonality of the FCD slope aggregated over the two regions of interest is not strongly affected by the different algorithms considered here.

Inferring sea ice properties with along-track ICESat-2 altimetry is subject to several caveats. The use of the FCD to approximate the FSD can lead to a biased interpretation of floe sizes pertaining to the alignment between the satellite tracks and sea ice leads (Horvat et al., 2019; Farrell et al., 2020). Further work, such as comparisons with overlapping imagery, would be necessary to assess whether the floe chord estimates used in this work can adequately capture the anisotropy of the pack at the scale of the perennial sea ice cover (Koo et al., 2023). Moreover, accurately estimating the location of leads along ICESat-2



tracks is an ongoing challenge, notably for dark leads (Petty et al., 2021). While quantitative estimates of the FCD, LWD and floe roundness vary with the different lead definitions considered in this work, our qualitative conclusions are robust to this choice. Furthermore, our use of a single slope to characterize the seasonal evolution of the FCD, fITD and LWD distributions can blur potentially different mechanisms occurring across a wide range of scales. Future work may benefit from considering

alternative approximations of these distributions to explore more detailed and localized processes within the pack (Montiel and Mokus, 2022). Finally, ICESat-2 freeboard estimates do not allow us to parse the contributions of snow versus ice, which limits our ability to understand the mechanisms responsible for the observed seasonality in fITD and floe roundness. Conducting floe-scale analyses using a snow-penetrating altimeter could be useful extension to this work.

## 5    Conclusions

This study uses various satellite products, including along-track altimetry from ICESat-2, to explore the seasonality of the perennial sea ice pack in the Weddell Sea. We contrast the behavior of the western portion of the pack, characterized by a high fraction of MYI, to the southern region, which is largely composed of FYI. Despite different sea ice types, the seasonality of the FCD, a proxy for the FSD, is consistent between the two regions. On the other hand, the seasonality of the fITD suggests an anti-phase relationship between the two regions, potentially due to the differential effects of thermodynamics versus dynamics

in controlling ice thickness within the two regions. These results show that regional differences in ice concentration and type at larger scales occur in conjunction with different behaviours at the small scale. Further, similarities in the seasonality of the floe chord distribution and lead width distribution between the two regions studied here do not translate to the same seasonality freeboard ice thickness distribution, vertical roundness, or lead width distribution.

The mean vertical freeboard of floes exhibits a dome-shaped profile, characterized by thinner ice at the floe edge and thicker

ice towards its center. Composited profiles show that larger floes tend to be thicker and more vertically square than smaller floes. The proliferation of smaller and vertically rounder floes in the summer causes an increase in the mean vertical roundness of floes in the southern portion of the sea ice pack. The mean spacing between leads is smaller in the west compared to the south, which is consistent with a more compact sea ice cover in the west, likely due to compaction against the Antarctic Peninsula (Kacimi and Kwok, 2020).

Feedbacks linking basin to floe-scale processes are not resolved in most climate models and may in part explain biases in modelled sea ice trends. The sea ice metrics considered here may help calibrate floe-resolving models (Manucharyan and Montemuro, 2022; Moncada et al., 2023) and facilitate the development of subgrid parameterizations designed for continuum-based sea ice models (Roach et al., 2018). The roundness of the dome-shaped vertical profiles exhibited by floes may notably help parameterize lateral erosive processes, which likely play an important role in shaping the joint floe chord and freeboard

thickness distribution. The freeboard thickness profiles may help examine the effects of lateral melt (Horvat et al., 2016; Gupta and Thompson, 2022), flooding (Ackley et al., 2020) and snow export (Leonard and Maksym, 2011; Moon et al., 2019), but disentangling their relative importance require independent estimates of snow and ice thickness over individual floes. We





therefore suggest that a comparison of such metrics from ICESat-2 in a region with upcoming floe-scale models could provide a useful diagnostic of model performance.

As basin-wide forcings continue to change around Antarctica, austral sea ice may transition towards a pack composed of smaller and thinner floes that are more susceptible to drift and melt. Evidence for a younger, thinner and more mobile sea ice cover has been reported in the Arctic ocean (Rampal et al., 2009; Notz and Stroeve, 2016; Mallett et al., 2021), while Antarctica has experienced several recent episodes of rapid sea ice loss, suggesting a potential regime shift towards longer term decline (Turner et al., 2022; Purich and Doddridge, 2023). Finely-resolved inferences on the winter to summer transition

of the pack, such as the ones presented in this work, will help improve our understanding of longer term warming trends and more carefully parse regional differences than smoothed satellite products. These prospects thus motivate the need for continued sea ice monitoring at the floe scale.

*Data availability.* The ICESat-2 data can be accessed at https://nsidc.org/data/icesat-2/data, the sea ice type data at https://seaice.uni-bremen. de/data/MultiYearIce and the Sentinel-2 images at https://sentinel.esa.int/web/sentinel/user-guides/sentinel-2-msi/revisit-coverage.



**Appendix A**

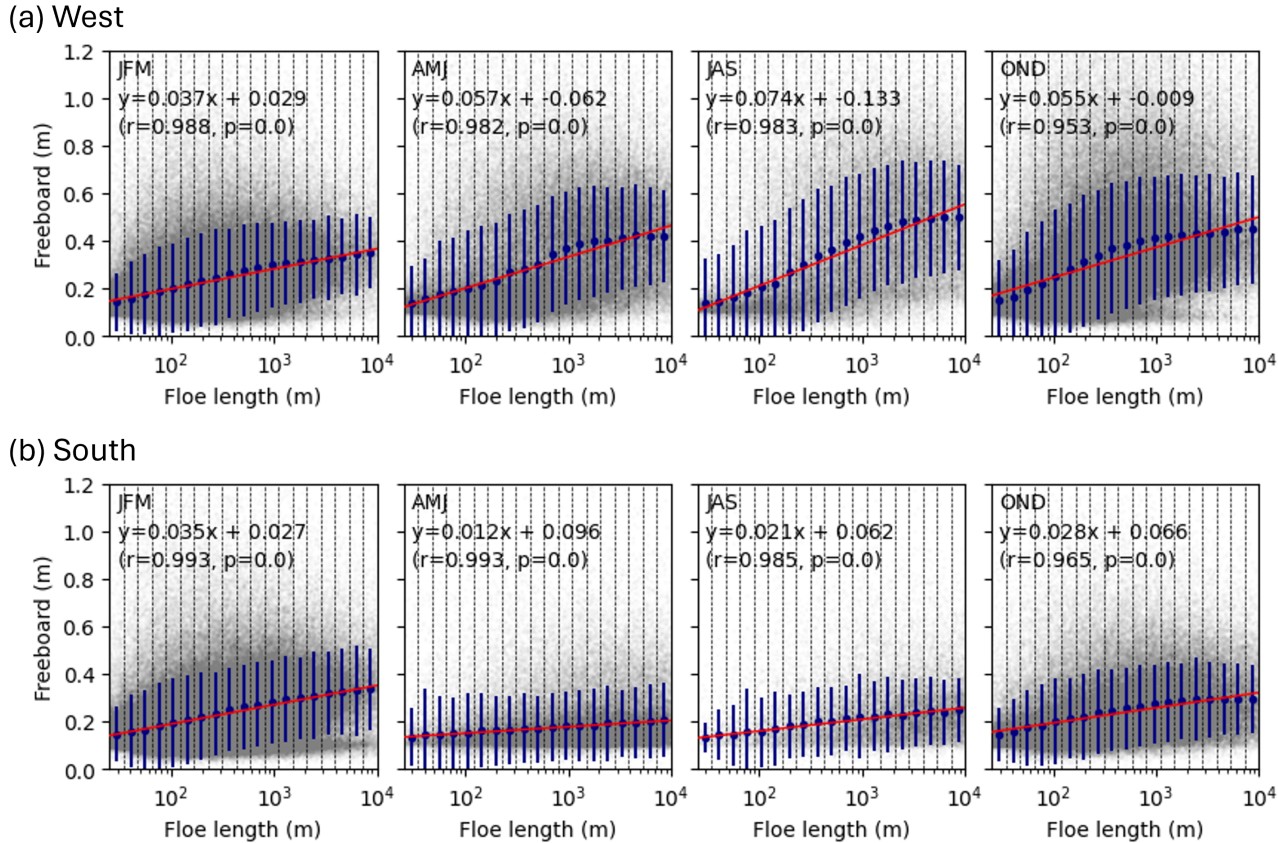

**Figure A1.** Mean floe freeboard versus floe chord length aggregated in four seasons during the years 2019 to 2022 for the (a) western and (b) southern regions of the pack. The floe length is divided into 20 ranges on a log scale. The red line indicates the linear best fit line and its corresponding statistics are detailed in each panel. The navy dots represents the median freeboard for each bin range, and the vertical navy bar indicates the standard deviation of freeboard.



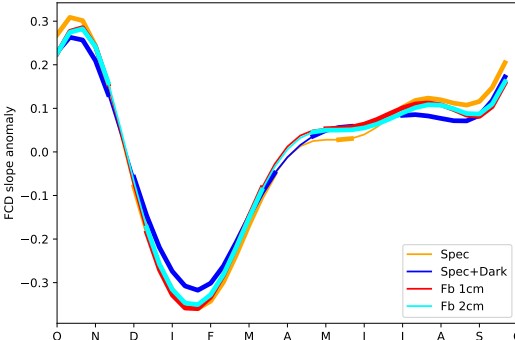

**Figure A2.** Seasonal composite of the FCD slope anomaly, calculated as in Fig. 4 (a), for the following lead detection methods: 'Specular', 'Specular + Dark', 'Freeboard height threshold at 2 cm' and 'Freeboard height threshold at 2 cm'. The thickner parts of each curve are significantly different from zero, based on a p-value of 0.05.

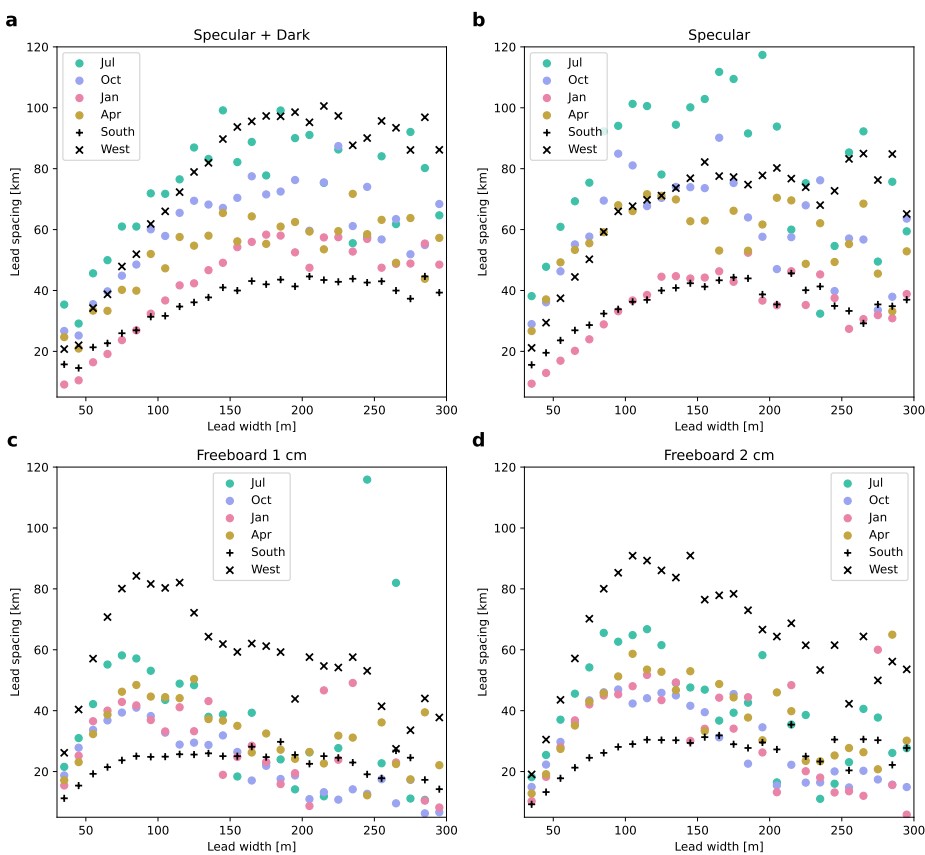

**Figure A3.** Sensitivity of the lead spacing seasonality to lead definitions for the combined western and southern regions. The analysis is conducted as in Fig. 6 The lead definitions are detailed in the caption of Fig. A2.



*Author contributions.* M.G., H.R., Y.K., S.M.T.C, X.L. and P.H. designed the study; M.G., H.R., Y.K. and S.M.T.C processed the data; all authors analyzed the results. All authors contributed to and reviewed the manuscript.

*Competing interests.* At least one of the (co-)authors is a member of the editorial board of The Cryosphere.

*Acknowledgements.* MG was supported by the Office of Naval Research (N00014-19-1-242) initiative on Mathematics and Data Science
for Physical Modeling and Prediction of Sea Ice. SC and PH acknowledge support from the Australian Government as part of the Antarctic Science Collaboration Initiative (ASCI000002), and were supported by the Australian Government's Australian Antarctic Science [AAS] Program grants 4506 & 4625, and by International Space Science Institute award #501. PH was supported by AAS grants 4496 & 4593.



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
