# Peer review of "Inferring the seasonality of sea ice floes in the Weddell Sea using ICESat-2"

_EGUsphere, 2024_

## Referee Comment (RC1)

**Inferring the seasonality of sea ice floes in theWeddell Sea using ICESat-2**

Mukund Gupta, Heather Regan, YoungHyun Koo, Sean Minhui Tashi Chua, Xueke Li, and Petra Heil

**23 June 2024**

**General comments**

The authors have conducted a study which is related to a quantitative analysis of sea ice floe characteristics in the Weddell Sea using ICESat-2 high-resolution altimetry data. The study focused on two areas of the Weddell Sea: the western and southern areas, which exhibit different sea ice conditions (as shown in Figure 2). The authors used the MICIT (Multiyear Ice Concentration and Ice Type) dataset to estimate daily concentrations of multiyear ice (MYI), first-year ice(FYI), young ice (YI) and open water, and derieved floe chord distribution (FCD), ice thickness distribution (fITD), lead width distribution(LWD) and vertical floe roundness from ICESat-2 along-track ATL10 data. The main insights the study offered about the seasonal variations in these characteristics were as follows:

1. the seasonality of the FCD is consistent, coinciding with the asymmetric melt/freeze cycle of the pack, while the seasonality of the fITD suggests an anti-phase relationship between the two regions.

2. The LWD is almost identical for the two regions, characterized by a mostly monotonic decrease with size, while the mean lead spacing in the west is up to 2.5 times larger than that in the south.

3. The vertical floe roundness by Compositing floe profiles shows that smaller floes are more vertically round than larger floes, and that the mean roundness of floes increases during the melt season.

This work is justified, as the authors state, konwledge of floe-scale charactistics is important for understanding the sea ice dynamic. However, I think there are some room for improvements. It should be a good contribution to the TC and sea ice communities, after comments here or from the other reviews are adequately resolved.

**specific comments**

**1. Introduction**

Page 1, line 44: "Due to the sparseness of polar imagery, inter-regional comparisons of floe-scale properties are also limited..."

Although there are difficulties in collecting imagery, it would be better to mention the efforts they have made (Koo et al., 2023; Muchow et al., 2021). Studying sea ice in the Weddell using a combination of imagery and altimetry seems feasible.

**2. Data products**

Or Datasets and method?

Page 4, line 89: "We use the uncorrected version of the dataset, as the corrected version does not provide FYI and YI concentrations, though the timeseries of multiyear ice (MYI) area in theWeddell Sea compares well between the two products (not shown)."

Why don't you use the version with temperature and drift correction? Although the corrected version does not distinguish between the FYI and YI, but classifies them as non-MYI, it should provide a more reliable distribution of MYI and non-MYI, which is more important. Or it would be better to show the differences between the two versions in the appendix.

Page 4 108: "we define the ice located between two consecutive leads along a track as a single floe, and the extentof that ice segment as the floe chord length (Figure 1)."

How do you deal with the situation often found in ATL07/ATL10 where not all the segments on a lead could be classified as a lead (Figure 8 in Koo et al., 2023)? I think it would increase the number of small floe chords by your current method.

Page 5 line 120: "We test the sensitivity of our results to the following different lead definitions: (i) Specular + Dark (default)..."

In ATL07, there is actually a third lead definition, ssh_flag, which is the result of the radiometric decision tree and local height filter. Can you at this point add this lead

sensitivity test if the "ssh_flag" also exists in ATL10?

**3. Results**

Page 6 line 135: "the south and west (blue and yellow boxes in Figure 2)."

The southern region should be in the red box. The same typo is also in the description of Figure 2.

Page 6 line 138: "The melt/freeze cycle is asymmetric, characterized by rapid melt between November and February, with approximately 15-30 % loss in freeboard thickness (except in 2020-2021)."

Is there any explanation for the anomalous increase in sea ice concentration in January? Or a data problem?

Page 6 line 147: "Melsheimer et al. (2023)."

Should be (Melsheimer et al., 2023).

Page 6 line 147: "Between July and December, the total sea ice concentration in the western region tends to drop, driven almost exclusively by a decline in the MYI concentration."

Why the total sea ice concentration tends to decrease between July and December in both regions, while the freeboard thickness shows different trends in 2020 and 2021.

Page 8 line 174: "Very high freeboard values (> 1 m) are also observed over small floe lengths, which may be affected by the presence of icebergs, as discussed in Section 2.3."

I think it might also be affected by misclassification of real sea ice as lead. For example, if there are several sea ice segments on a thick floe that are classified as lead, they will be calculated as some small floes with high freeboard.

Page 10 figure 4c: "$\alpha'_{FCL}$"

Should be $\alpha'_{CLF}$.

Page 11 line 212: "Lead width and spacing."

A little confused about the definition of the lead width and the lead spacing. I cannot find a clear answer. Are they the total length of consecutive leads and the distance between two leads, saparetely? The lead spacing looks like the floe chord. It would be better to clarify them here or in the section 2.3.

Page 13 figure 6: "while the black 'plus' and 'cross' symbols represent yearly-mean aggregates for the two regions, respectively."

There is no the black 'plus' symbol in the figure.

Koo, Y., Xie, H., Kurtz, N.T., Ackley, S.F., Wang, W., 2023. Sea ice surface type classification of ICESat-2 ATL07 data by using data-driven machine learning model: Ross Sea, Antarctic as an example. Remote Sensing of Environment 296, 113726. https://doi.org/10.1016/j.rse.2023.113726

Muchow, M., Schmitt, A.U., Kaleschke, L., 2021. A lead-width distribution for Antarctic sea ice: a case study for the Weddell Sea with high-resolution Sentinel-2 images. The Cryosphere 15, 4527–4537. https://doi.org/10.5194/tc-15-4527-2021

---

## Referee Comment (RC2)

**General Comments**

The study provides valuable insights into the seasonal dynamics of sea ice floes in the Weddell Sea using high-resolution data from ICESat-2. The use of altimetry to quantify floe chord and freeboard thickness distributions, as well as lead width and vertical floe roundness, adds a crucial dimension to understanding sea ice behavior and its regional variations. This study contributes to the development of floe-resolving models by offering detailed diagnostics of sea ice processes.

However, I have several concerns regarding the usage of the ICESat-2 datasets and the clarity of the methodology. Therefore, I recommend that the paper undergo major revisions before it can be considered for publication.

**Here are my major comments:**

1. I'm not sure if the manuscript presents enough innovative methodologies/findings since many previous publications have already used altimeters, including CryoSat-2 and EnviSat, to derive floe chord distribution (Horvat et al., 2019) and lead-to-floe (Tilling et al., 2019). Additionally, it is unclear whether the results might depend on the specific ATL 07/10 version.
2. The introduction mentions that "The perennial extent of Antarctic sea ice is small compared to the seasonal portion of the pack…" and the manuscript primarily addresses perennial ice. Clarify how the study's focus on perennial ice informs the basin-wide behavior of the pack as stated in Line 74.
3. I don't quite follow the application of floe-derived metrics for model diagnostic evaluation since ICESat-2 and freeboard ice thickness distribution here is snow freeboard, not directly the ice thickness or ice freeboard. This means the snow freeboard needs to be converted with snow information to have potential sea ice process applications. It's unclear how we can use this knowledge—is it just a product-based ice diagnostic study?
4. I'm not clear how you define distributions such as floe chord distribution or freeboard ice thickness distribution in which temporal or spatial windows. While the floe chord length is defined by the distance between ice segments by each beam, what exactly is the floe chord distribution?
5. I appreciate the use of different methods in lead detection from ICESat-2 for the sensitivity test, but since the sensitivities are all based on ICESat-2 data, how about using one case to show the lead bias or validate the lead detection from a different data source, such as SAR? How does the systematic bias in lead detection affect the distribution slope changes in the results?

**Detailed comments:**

1. Line 89: What corrections were made in the previously uncorrected ice type product?
2. Line 93: How did you complement the daily ice type with monthly ICESat-2 and weekly ice motion data? More details are needed here.
3. Figure 2: Are Figures 2c and 2d calculated based on the Weddell Sea or the Antarctic basin scale?
4. Figure 3: Are the results here based on all seasons during the period of 2018-2022?

5. Line 194: It might not be feasible to describe this as "inter-annual variability" given it is only four years of data. The sample is scarce in terms of defining inter-annual variability.
6. Line 196: Based on Figure 4(b), is there statistically significant anti-correlation? Only sometimes in January and March do they share significant correlation instead of the whole season. How do you explain this anti-correlation?
7. Line 254: How can we trust the lead spacing from lead detection based on ICESat-2? Figure A3 shows huge differences in those spacings from different algorithms, especially over the west region. How does this affect the results in Figure 5b?
8. Figure 6: Where are the 'plus' symbols in the plot?
9. Equation (1): What do $d\hat{x}$ and $\hat{p}(\hat{x})d\hat{x}$ mean, and what is the temporal/spatial scale you used to derive the vertical roundness values?
10. Figure 7b: I'm curious about how to interpret the differences between the west and south regions in their features of vertical roundness.
11. Line 300: Which basin-scale are you referring to: the Weddell Sea or the Antarctic basin?
12. Figure A2: Should be "'Freeboard height threshold at 1 cm' and 'Freeboard height threshold at 2 cm'."

**Reference**

Horvat, C., Roach, L.A., Tilling, R., Bitz, C.M., Fox-Kemper, B., Guider, C., Hill, K., Ridout, A. and Shepherd, A., 2019. Estimating the sea ice floe size distribution using satellite altimetry: theory, climatology, and model comparison. The Cryosphere, 13(11), pp.2869-2885.

Tilling, R., Ridout, A. and Shepherd, A., 2019. Assessing the impact of lead and floe sampling on Arctic sea ice thickness estimates from Envisat and CryoSat-2. Journal of Geophysical Research: Oceans, 124(11), pp.7473-7485.

---

## Author Comment (AC1)

General comments

The authors have conducted a study which is related to a quantitative analysis of sea ice floe characteristics in the Weddell Sea using ICESat-2 high-resolution altimetry data. The study focused on two areas of the Weddell Sea: the western and southern areas, which exhibit different sea ice conditions (as shown in Figure 2). The authors used the MICIT (Multiyear Ice Concentration and Ice Type) dataset to estimate daily concentrations of multiyear ice (MYI), first-year ice(FYI), young ice (YI) and open water, and derieved floe chord distribution (FCD), ice thickness distribution (fITD), lead width distribution(LWD) and vertical floe roundness from ICESat-2 along-track ATL10 data. The main insights the study offered about the seasonal variations in these characteristics were as follows:

1. the seasonality of the FCD is consistent, coinciding with the asymmetric melt/freeze cycle of the pack, while the seasonality of the fITD suggests an anti-phase relationship between the two regions.
2. The LWD is almost identical for the two regions, characterized by a mostly monotonic decrease with size, while the mean lead spacing in the west is up to 2.5 times larger than that in the south.
3. The vertical floe roundness by Compositing floe profiles shows that smaller floes are more vertically round than larger floes, and that the mean roundness of floes increases during the melt season.
This work is justified, as the authors state, knowledge of floe-scale chrematistics is important for understanding the sea ice dynamic. However, I think there are some room for improvements. It should be a good contribution to the TC and sea ice communities, after comments here or from the other reviews are adequately resolved.

Thank you for your review and for recognizing the contributions made by our work.

Specific comments

1. Introduction
Page 1, line 44: "Due to the sparseness of polar imagery, inter-regional comparisons of floe-scale properties are also limited..."
Although there are difficulties in collecting imagery, it would be better to mention the efforts they have made (Koo et al., 2023; Muchow et al., 2021). Studying sea ice in the Weddell using a combination of imagery and altimetry seems feasible.

Agreed. We have added a sentence to that effect on L44-46.

2. Data products
Or Datasets and method?

We have changed to 'Datasets and methods'
Page 4, line 89: "We use the uncorrected version of the dataset, as the corrected version does not provide FYI and YI concentrations, though the timeseries of multiyear ice (MYI) area in the Weddell Sea compares well between the two products (not shown)."

Why don't you use the version with temperature and drift correction? Although the corrected version does not distinguish between the FYI and YI, but classifies them as non-MYI, it should provide a more reliable distribution of MYI and non-MYI, which is more important. Or it would be better to show the differences between the two versions in the appendix.

We now provide a comparison between the corrected and uncorrected versions of the MYI product in Fig. A1. This figure shows that the two products compare well in the two regions of interest. We keep the uncorrected version in the main paper because the corrected version does not provide data between November and March, and because it does not provide further decomposition into FYI and YI.

Page 4 108: "we define the ice located between two consecutive leads along a track as a single floe, and the extent of that ice segment as the floe chord length (Figure 1)."
How do you deal with the situation often found in ATL07/ATL10 where not all the segments on a lead could be classified as a lead (Figure 8 in Koo et al., 2023)? I think it would increase the number of small floe chords by your current method.

This issue is indeed a limitation that is common to all the studies estimating FSD from altimetry. As shown in Koo et al., 2023, different lead definitions have different sensitivity in capturing leads. In some cases, leads can be misclassified as sea ice, whereas the opposite may occur in other cases. We find here that the absolute value of the FCD slopes changes with the different lead definitions considered (Fig. A2), but the seasonality is robust across them, which is what our study focuses on.

We have made a note about the possibility of sea ice misclassified as leads on lines 125-126 and 364-365.

Page 5 line 120: "We test the sensitivity of our results to the following different lead definitions: (i) Specular + Dark (default)..."
In ATL07, there is actually a third lead definition, ssh_flag, which is the result of the radiometric decision tree and local height filter. Can you at this point add this lead sensitivity test if the "ssh_flag" also exists in ATL10?

We have included this definition in Fig A2 and Fig A3.

3. Results

Page 6 line 135: "the south and west (blue and yellow boxes in Figure 2)."
The southern region should be in the red box. The same typo is also in the description of Figure 2.

Corrected

Page 6 line 138: "The melt/freeze cycle is asymmetric, characterized by rapid melt between November and February, with approximately 15-30 % loss in freeboard thickness (except in 2020-2021)." Is there any explanation for the anomalous increase in sea ice concentration in January? Or a data problem?

This is indeed an interesting feature. By inspecting maps of sea ice concentration over time, it seems like this increase in sea ice concentration in the South Box might be associated with increased advection of high-concentration ice from the East. However, given the relatively small magnitude of the increase signal, more work would be needed to understand this effect and assess its statistical significance over the years.

Page 6 line 147: "Melsheimer et al. (2023)."
Should be (Melsheimer et al., 2023).

Corrected.

Page 6 line 147: "Between July and December, the total sea ice concentration in the western region tends to drop, driven almost exclusively by a decline in the MYI concentration."
Why the total sea ice concentration tends to decrease between July and December in both regions, while the freeboard thickness shows different trends in 2020 and 2021.

In the western region, the total sea ice concentration and average freeboard thickness evolve mostly in phase (thinner and less concentrated ice during the melt season), which matches our general expectation.

In the southern region, the total sea ice concentration and average freeboard thickness are not always in phase, indeed. It should be noted that the magnitude of the seasonal cycle in these two quantities is generally weaker in the south than in the west. Therefore, the fluctuations in the south are likely to more strongly reflect longer term variability (e.g. inter-annual). These changes are more difficult to interpret, especially over the 4-year period that we consider.

The slight decline in concentration between January and October in the south happens every year and can be interpreted as net export of sea ice from that region dominating over areal growth. Over most years (except 2019), the freeboard thickness tends to increase between February and October, which reflects thermodynamic growth. In 2019, the freeboard thickness decreases slightly during between January and October, which may be caused by net export of thicker ice. We note however that the relatively sparseness of the freeboard thickness data used to generate the ATL20 product or changes in snowfall could affect some of these inferences.

We have added some more explanations about the seasonal behavior of the total sea ice concentration and mean freeboard thickness on Lines 134-166.

Page 8 line 174: "Very high freeboard values (> 1 m) are also observed over small floe lengths, which may be affected by the presence of icebergs, as discussed in Section 2.3."
I think it might also be affected by misclassification of real sea ice as lead. For example, if there are several sea ice segments on a thick floe that are classified as lead, they will be calculated as some small floes with high freeboard.

We agree that misclassification of sea ice as leads may occur and have added a statement to that effect on Line 124-127: 'We also note that the ATL07/10 lead detection product does not always

capture leads that are visible from concomitant Sentinel-2 imagery (Koo et al. 2023), and may erroneously classify certain leads as sea ice. Additionally, some sea ice segments may be mistaken for leads, particularly within ICESat-2's `dark lead' classification.'

Nevertheless, we do not have evidence that the misclassification of sea ice as leads occurs more frequently than the opposite, or more frequently within thicker sea ice. Therefore, we do not further highlight that caveat when discussing small and thick floes on Lines 188-189.

Page 10 figure 4c: "αLCF' "
Should be αCLF'.

Corrected.

Page 11 line 212: "Lead width and spacing."
A little confused about the definition of the lead width and the lead spacing. I cannot find a clear answer. Are they the total length of consecutive leads and the distance between two leads, separately? The lead spacing looks like the floe chord. It would be better to clarify them here or in the section 2.3.

- The lead width is the distance between the first and last of segments identified as a single contiguous lead along a track.
- The lead spacing is calculated here between leads that are within the same lead width bin (as defined in Fig 6), so it is different to the floe chord.

We have included these clarifications on Lines 227-228 and Lines 247-248, respectively.

Page 13 figure 6: "while the black 'plus' and 'cross' symbols represent yearly-mean aggregates for the two regions, respectively."
There is no the black 'plus' symbol in the figure.

Corrected.

Koo, Y., Xie, H., Kurtz, N.T., Ackley, S.F., Wang, W., 2023. Sea ice surface type classification of ICESat-2 ATL07 data by using data-driven machine learning model: Ross Sea, Antarctic as an example. Remote Sensing of Environment 296, 113726. https://doi.org/10.1016/j.rse.2023.113726

Muchow, M., Schmitt, A.U., Kaleschke, L., 2021. A lead-width distribution for Antarctic sea ice: a case study for the Weddell Sea with high-resolution Sentinel-2 images. The Cryosphere 15, 4527–4537. https://doi.org/10.5194/tc-15-4527-202

---

## Author Comment (AC2)

**General Comments**

The study provides valuable insights into the seasonal dynamics of sea ice floes in the Weddell Sea using high-resolution data from ICESat-2. The use of altimetry to quantify floe chord and freeboard thickness distributions, as well as lead width and vertical floe roundness, adds a crucial dimension to understanding sea ice behavior and its regional variations. This study contributes to the development of floe-resolving models by offering detailed diagnostics of sea ice processes. However, I have several concerns regarding the usage of the ICESat-2 datasets and the clarity of the methodology. Therefore, I recommend that the paper undergo major revisions before it can be considered for publication.

Thank you for your review and for recognizing the advances made by our work.

**Here are my major comments:**

1.  I'm not sure if the manuscript presents enough innovative methodologies/findings since many previous publications have already used altimeters, including CryoSat-2 and EnviSat, to derive floe chord distribution (Horvat et al., 2019) and lead-to-floe (Tilling et al., 2019). Additionally, it is unclear whether the results might depend on the specific ATL 07/10 version.

We certainly acknowledge that several past studies have developed methodologies and estimated floe characteristics from altimetry. For example, Horvat et al. (2019) explored the spatial and temporal changes in FCD for the Arctic Ocean using CryoSat-2; Tilling et al. (2019) used CryoSat-2 to help validate the coarser measurements of Envisat in the Arctic Ocean for floe size and thickness; Petty et al. (2021) assessed the ability to use ICESat-2 to extract floe and lead characteristics in both hemispheres; Farrell et al. (2020) discussed the topography of sea ice using ICESat-2. We cite these papers and others in the introduction.

Here, our primary focus was to better understand the seasonal cycle of sea ice within the perennial ice pack of the Weddell Sea at finer scales than has previously been done, and connect these fine scales inferences to the larger-scale behavior of the pack, as inferred from coarser satellite measurements. In the process of studying the behavior of floes and leads within this region, we developed some novel metrics (e.g. floe roundness and lead spacing) that we argue are useful in characterizing the properties of the pack, and could therefore be applied to other regions in future work. It was nevertheless not our main intent to develop new methodologies for processing altimeter data.

Our main novel scientific findings are as follows:

1.  The seasonality of the FCD is tied to the melt/freeze cycle, in both the west and south Weddell Sea, despite their substantial differences in ice types.

2. By contrast, the seasonality of the fITD suggests mirrored behavior during the melt season despite the total freeboard thickness having a similar phasing between these two regions.

3. There is a positive correlation between sea ice freeboard thickness and floe chord length (ie. larger floes tend to be thicker). This correlation is observed throughout the year.

4. Smaller floes are rounder than larger ones. The mean roundness of floes increases during the melt season.

We believe these results deepen our understanding of sea ice behavior at fine scales and over the seasonal cycle. This new understanding could then help test and calibrate floe-aware sea ice models.

We have modified the first paragraph of the discussion to reflect the primary focus of our work on building scientific understanding rather than the systematic development of new methodologies. Moreover, the key points above are summarized in the abstract and the conclusions.

Regarding the ICESat-2 version, we use the latest version of the ATL07/10 product (v.6), as it includes the latest processing algorithms published by the ICESat-2 team. In an earlier iteration of the paper, we had used v.5, but since that version has now been supplanted by v.6, we choose to focus on the latter. Note that our main results did not change substantially when switching from v.5 to v.6 (see results below for FCD and fITD). It is beyond the scope of this study to perform a comprehensive assessment across the various versions of the ICESat-2 product .

[Figure]

2.  The introduction mentions that "The perennial extent of Antarctic sea ice is small compared to the seasonal portion of the pack…" and the manuscript primarily addresses perennial ice. Clarify how the study's focus on perennial ice informs the basin-wide behavior of the pack as stated in Line 74.

We agree that our study only provides inferences within the perennial sea ice region. We have changed this line to:

 'This work uses the ICESat-2 altimeter product to examine the seasonality of the perennial sea ice zone in the Weddell Sea and explore the utility of floe-level metrics in interpreting the larger-scale behavior of the pack.'

3.  I don't quite follow the application of floe-derived metrics for model diagnostic evaluation since ICESat-2 and freeboard ice thickness distribution here is snow freeboard, not directly the ice thickness or ice freeboard. This means the snow freeboard needs to be converted with snow information to have potential sea ice process applications. It's unclear how we can use this knowledge—is it just a product-based ice diagnostic study?

The freeboard information provided by ICESat-2 includes the snow layer and ice above sea level. We acknowledge the limitation of not separating the contributions between snow and ice (as detailed within the Discussion and the Conclusions sections of the paper). Performing this separation is difficult at the fine scales considered here, because it requires high-resolution information about snowfall, compaction, flooding and redistribution by winds, which is currently not available at the floe scale. We therefore focus purely on freeboard thickness to avoid introducing biases related to these uncertain snow-related processes. This is still a useful quantity, since it has been shown that data assimilation and model validation can be performed with sea ice freeboard (see for example Sievers et al., 2023). It would be the subject of follow-up work to examine snow versus ice contributions, by combining ICESat-2 with other datasets. We have added a note about the potential for data assimilation on Line 400.

Moreover, the inferences in this work pertaining to the LWD, FCD and lead spacing do not depend on conversion between snow and ice fractions.

[Sievers, I., Rasmussen, T. A., & Stenseng, L. (2023). Assimilating CryoSat-2 freeboard to improve Arctic sea ice thickness estimates. The Cryosphere Discussions, 2023, 1-23.]

4.  I'm not clear how you define distributions such as floe chord distribution or freeboard ice thickness distribution in which temporal or spatial windows. While the floe chord length is defined by the distance between ice segments by each beam, what exactly is the floe chord distribution?

The FCD and fITD are calculated over time windows (chunks) of 3 days, and aggregated between all the floes detected within each region (south and west), respectively. We consider the full time

period (Oct 2018-2022). Note that we tried chunks, varying from 2 to 10 days, and did not find substantial differences in our results. We have added the relevant information on Lines 199-203.

A floe chord is defined as the distance between two consecutive lead edges. The FCD is defined as the number of floes binned over chord lengths. We evaluate it for each region separately. We have added the following text on Line 172:

'The FCD is defined as the count of individual floes binned over their respective chord lengths (Fig. 3 (a)). Aggregated over the full study period (October 2018 - October 2022), …'

And the following on Line 178-179.

'The fITD is evaluated as the count of individual ICESat-2 segments binned over their respective freeboard thickness (Fig. 3 (b)). Aggregating data over the full study period, …'

5.  I appreciate the use of different methods in lead detection from ICESat-2 for the sensitivity test, but since the sensitivities are all based on ICESat-2 data, how about using one case to show the lead bias or validate the lead detection from a different data source, such as SAR? How does the systematic bias in lead detection affect the distribution slope changes in the results?

Comparisons between floe chord lengths derived from ICESat-2 and imagery have been performed by several studies, including Koo et al. (2023), Petty et al. (2021) and Farrell et al. (2020). Koo et al. (2023) investigate the effects of the different lead detection algorithms used in this work, and place them in the context of leads derived from imagery in the Ross Sea. The authors find that different techniques work better for different scenarios, and that there is no one technique that always results in more accurate classification than the others. In some instances, sea ice is misclassified as a lead, while in others the opposite occurs. In some cases, the distinction may actually be ambiguous from the imagery itself, especially in the presence of thin nilas. Given that our study spans several sea ice regimes, it is difficult to robustly establish the sign and magnitude of any potential systematic, as those could vary in time and in space. Performing an additional single case study would not allow us to quantify the biases over all the regimes we consider here, as it would necessarily be limited in time and space.

Instead, performing a thorough analysis regarding the seasonal evolution of FCD and LWD using SAR imagery and comparing with our results would be valuable. However, this would require a considerable number of images, taken over the full span of the study period and region, along with a validated floe segmentation algorithm and a robust assessment of uncertainties. This would be a separate study in itself and is beyond the scope of this work.

**Detailed comments:**

1.  Line 89: What corrections were made in the previously uncorrected ice type product?

The corrections pertain to the MYI concentration, as described in Melsheimer et al. (2023). We did not perform these corrections ourselves - instead they were performed as part of the study in Melsheimer et al. (2023), from which we sourced the MICIT data used in our work.

In brief, Melsheimer et al. (2023) apply two types of correction to the MYI concentration: (1) temperature-based, and (2) drift-based. The temperature correction considers the fact that melting may lead to some MYI appearing as FYI in its scattering properties. The correction reclassifies FYI to MYI based on whether warm enough surface air temperatures were observed in the satellite record. The drift correction uses the fact that by definition, no new MYI can be generated after the end of the melt season. Therefore, MYI should only be found within regions where it may have realistically drifted after that period. The drift correction uses a sea ice drift product to delimit these regions, and reclassify MYI outside these regions as non-MYI.

In the paper, we now provide a comparison between the corrected and uncorrected versions of the MYI product in Fig. A1. This figure shows that the two products compare well in the two regions of interest. We keep the uncorrected version in the main paper because the corrected version does not provide data between November and March, and because it does not provide further decomposition into FYI and YI.

2. Line 93: How did you complement the daily ice type with monthly ICESat-2 and weekly ice motion data? More details are needed here.

Apologies for the confusing phrasing. We use ice type data, ICESat-2 and motion data, separately. We did not combine them. We have now rephrased this to: 'Additionally, we use gridded sea ice freeboard…'

3. Figure 2: Are Figures 2c and 2d calculated based on the Weddell Sea or the Antarctic basin scale?

They are calculated based on the South and West regions shown in Fig. 2a and b. This is specified in the caption.

4. Figure 3: Are the results here based on all seasons during the period of 2018-2022?

Yes. We have added this line to the caption of Fig. 3: 'Floe-scale properties aggregated over all the ICESat-2 data collected between October 2018 and October 2022.'

5. Line 194: It might not be feasible to describe this as "inter-annual variability" given it is only four years of data. The sample is scarce in terms of defining inter-annual variability.

We removed the term 'inter-annual variability' and replaced it with 'the variability over the four years considered' or simply 'variability' throughout the text.

6. Line 196: Based on Figure 4(b), is there statistically significant anti-correlation? Only sometimes in January and March do they share significant correlation instead of the whole season. How do you explain this anti-correlation?

We agree that we cannot statistically determine whether alpha_ITD' is anti-correlated between the two regions for the seasonal cycle, especially as the signal in the west is only significant between February to August approximately. We have changed the phrasing to:

'Unlike the FCD, the seasonality of the fITD slope is not consistently in phase between the southern and western regions of the sea ice cover (Fig. 4 (b)).'

In the discussion, we suggest that the potential mirrored behavior may be due to the differential effects of thermodynamics versus dynamics in controlling ice thickness within the two regions. This would however need to be assessed with more observational data in the coming years or using model.

7. Line 254: How can we trust the lead spacing from lead detection based on ICESat-2? Figure A3 shows huge differences in those spacings from different algorithms, especially over the west region. How does this affect the results in Figure 5b?

There are indeed differences in the absolute values of lead spacing for different lead definitions. Nevertheless, the following patterns are consistent between them:
-   The lead spacing is generally smaller in the south than in the west
-   The lead spacing tends to decrease between July and October, increase from January to April and increase from April to July.

These are the points we comment on within the text (Section 3.3). There, we write the following regarding the sensitivity to lead definitions:

'Characterizing leads using a freeboard threshold instead of the identification provided by ICESat-2 can reduce the seasonality in the lead width spacing signal (Fig. A3). This may be due to the misidentification of thin ice as leads, especially near areas of widely distributed thin ice (Koo et al. 2023). Nevertheless, the seasonal trend in lead width spacing remains consistent across lead definitions.'

Regarding the effect of lead definitions on Fig. 5b, we have added a panel to Fig A2 that investigates the sensitivity of alpha_LWD' to the lead definition. We find that the phasing does differ with the different lead definitions, but broadly alpha_LWD' tends to be negative between December and February and positive between May and July.

We made a note of this on Lines 244-245.

8. Figure 6: Where are the 'plus' symbols in the plot?

This was a typo. We have fixed it.

9. Equation (1): What do $d\hat{x}$ and $\hat{p}(\hat{x})d\hat{x}$ mean, and what is the temporal/spatial scale you used to derive the vertical roundness values?

$\hat{x}$ is the normalized distance along a floe chord and $\hat{p}(\hat{x})$ is the corresponding freeboard height profile along the floe. We have added that information after Eq. (1).

In Fig 7a, we consider all the floes across both regions and all months to derive the roundness. In Fig 7b, we consider each month and the regions individually. This is specified in the corresponding caption.

10. Figure 7b: I'm curious about how to interpret the differences between the west and south regions in their features of vertical roundness.

It is difficult to thoroughly assess the differences between the two regions here because the seasonal variations in vertical roundness for the western regions remains within the year-to-year variability. This would be something to consider for future work with more years of data.

11. Line 300: Which basin-scale are you referring to: the Weddell Sea or the Antarctic basin?

Apologies for the confusion. We meant the perennial sea ice pack. We have changed this to: 'In spite of these regional differences, the seasonality of the FCD is consistent between the western and southern portions of the pack and is in phase with the asymmetric melt/freeze cycle over the perennial sea ice pack (Section 3.2)'.

12. Figure A2: Should be "'Freeboard height threshold at 1 cm' and 'Freeboard height threshold at 2 cm'."

Corrected.

Reference

Horvat, C., Roach, L.A., Tilling, R., Bitz, C.M., Fox-Kemper, B., Guider, C., Hill, K., Ridout, A. and Shepherd, A., 2019. Estimating the sea ice floe size distribution using satellite altimetry: theory, climatology, and model comparison. The Cryosphere, 13(11), pp.2869-2885.

Tilling, R., Ridout, A. and Shepherd, A., 2019. Assessing the impact of lead and floe sampling on Arctic sea ice thickness estimates from Envisat and CryoSat-2. Journal of Geophysical Research: Oceans, 124(11), pp.7473-7485.

---

## Author Comment (AC3)

**Reviewer 3**

This is a review of Gupta et al (2024).

The work uses ICESat-2 altimetry to understand properties of the shape and size of sea ice floes in the Weddell Sea. Generally the paper is well-written and interesting - as a case study in how laser altimetry can be used to explore the properties of the sea ice surface. I think there are some methodological questions I am hoping to have addressed in a revised manuscript.

The authors do discuss this, but it is worth emphasizing. The use of radiometrically-defined "dark lead"s, remains a challenge with ICESat-2, because of a lack of confidence in their meaning. I would like to see your Fig. A3 as an anomaly rather than side-by-side plots to show the differences. Perhaps in addition, showing the histograms next to one another of all lead spacings and chord lengths. Chord/spacing measurements with IS-2 have often been confusing, as with the high along-track resolution it is very easy to chop a floe in half with a misclassification. It might be in the Weddell this is not an issue, so that's exciting! But more details here might be helpful.

We certainly notice differences in lead locations and widths when considering different lead definitions for a given track. When averaging over the large areas of the south and west Weddell Sea, these differences translate into some differences in the FCD. The plot below shows the annual-mean FCD curves for different lead definitions, as requested. We will include this in the Appendix. Despite these differences, we find that the seasonality of the FCD slope anomalies is mostly robust across them (Fig. 4a). We cannot comment on whether this is true only for the Weddell Sea or elsewhere.

[Figure]

The plot below can allow you to more easily compare the lead spacings of Fig. A3 for the annual-mean fields. We will include this in the Appendix.

[Figure]

More details are needed on the power law slopes. For example, how are you fitting alpha to the FCD? A treatment of this mathematically is in several places, including Virkar and Clauset (2014), but has to be done carefully. The use of binning can introduce spurious errors in the slope of such a distribution - see Stern et al 2018. The VC method is simple to apply and doesn't rely on the binning. The statistical tests examined there should be applied before discussing power-law fits (if they are not already used).

All the slopes alpha_XX are obtained by linear regression (ordinary least squares) over the binned data. We recognise that this method has been shown to cause biases in the estimation of the exponent in a power law model. Alternative methods, such as suggested in the comment above, assume that the distribution follows a power law and use this assumption to reduce biases, notably those caused by the tail end of the distribution, where data is usually more sparse.

Here, we can clearly see that the FCD is not distributed like a power law over the range of floe chords that we considered - ie. the curves are visibly not straight lines in log-log space (Fig. 3a). Given this result, we do not believe that a power-law is necessarily the best model to fit this distribution, and others may indeed provide a better fit. However, our objective here is to characterise the broad-scale changes in the distribution over a seasonal cycle. We thus seek a simple model that is easily interpretable (in terms of slope and intercept) and can fit the curve reasonably well. The power law model meets these characteristics, with R2 typically varying between 0.9 and 0.98. Since we do not assume that the original curve follows a perfect power law, we do not believe it would be appropriate to use more sophisticated methods for inferring alpha_FCD.

We acknowledge that this approach comes with caveats:
- We are reducing a curve to a straight line. This prevents us from distinguishing over which range of floe chords the changes are occurring throughout the seasonal

cycle. More detailed investigations could be possible with a more complex model, but this would be more difficult to interpret than a single slope, and is beyond the scope of our study.
- Biases at the tail end may still exist, but this is difficult to eliminate without making an assumption about the nature of the distribution, which we avoid.
- Biases due to bin spacing may remain. We do not find a notable difference between them in the range 75 to 300 m for the annual mean value. The default was 75m.

For completeness, we calculate power law fits of the FSD using the Maximum Likelihood method outlined in Virkar and Clauset (2014). The plot below shows these fits over the seasonal cycle, for the different lead detection methods, compared with the ordinary least squares method. The Maximum Likelihood method does make a substantial difference to the fit, but the overall seasonality remains robust.

[Figure]

On Fig. 3a we will show the power law fit lines. We will add a condensed version of the above to the discussion, which applies to other alpha values as well.

I would suggest renaming your alphas (one is a power law, others are not) because they correspond to different distributions.

This could be possible indeed, but we feel it would impede readability slightly. We will instead indicate more frequently which type of distribution we are referring to.

It is also important to discuss that there is fITD uncertainty, much like FCD uncertainty, because of the unknown misalignment of the laser with sea ice structure (in this case, ridges, say).

That is a good point. We will add a sentence to the discussion.

The extremely high correlation ($R^2 = 0.98$) between mean freeboard heights and mean chord length is striking, but somewhat concerning - do you think this is because thicker ice is easier to separate from open water? In that case, thicker ice means fewer missed

classifications means wider FCD. Have you examined this possibility? Can you explain why this correlation is so strong?

The explanation you provide could indeed be part of the correlation we identify. The other physical processes we discussed may also play a role. However, we cannot disentangle their individual contributions. We will add the mechanism you propose to the discussion.

Regarding the interpretation of the high correlation, we note that the $R^2$ metric was applied to the 'mean' freeboard values when binned across chord lengths, which is well represented with a linear fit across seasons and regions (blue points and red line in Fig. A4), giving a high $R^2$ value. However, there is a substantial number of floes that deviate from the mean freeboard (grey clouds of points in Fig. A4), as quantified by the standard deviation (blue bars in Fig. A4).

We will add the standard deviation to Fig. 3c to avoid any misinterpretation of the $R^2$ value.

For the floe roundness elements - an important question here is what is the intended usage of floe freeboard roundness information? I assume this is to infer something about the dynamical behavior of the floes contacting on another? This could be better justified because this metric, especially with a laser altimeter that has uneven sampling and only measures freeboard, is somewhat unclear. It would be useful to see exactly how ICESat-2 "sees" an individual floe rather than the composite normalized image of Figure 7. - because floe surfaces are very non-round (see your Figure 1!) Freeboard variability can be from changes in sea ice density, and "floe roundness" is realistically more closely compared to the surface roundness alone. How round the floe is depends on factors under the ocean surface that ICESat-2 can't examine here.
It would be helpful to discuss averaging here - I presume in the paragraph beginning on L264 you mean you average over all floes, not average along the floe. How much variance is there in the resulting plots shown in Figure 7? This could be a good visualization - showing the variability associated with your compositing.

The roundness we calculate indeed only informs us about the surface above sea level, and not about what happens under water. We will remove 'lateral melt' as a possible explanation.

Yes, we do mean average over all floes.

The variability in the mean profiles in Fig 7 is indeed very high and reflects many asperities seen on floes in Fig. 1. The shading in the plot below reflects 2 sigma standard deviation. This means that the roundness metric is unlikely to be useful when only considering a small number of them. Nevertheless, we find it interesting that the shape collapses to such clearly defined half domes when averaging over a sufficient number of floes. It remains to be determined what processes may be responsible for this. As our knowledge of floe-level

processes continues to increase, floe roundness may be a useful metric for differentiating the variability of these processes over time and spatial scales, and for model calibration.

Note that we also noticed a bug in our calculation for this plot where the curves were being scaled down artificially. We have now corrected this. Therefore, the d values shown below are slightly different to the submitted manuscript. We will add the corrected version of the plot to the manuscript, including the variability of the profiles.